# FROM ISOLATION TO ENTANGLEMENT: WHEN DO INTERPRETABILITY METHODS IDENTIFY AND DISENTANGLE KNOWN CONCEPTS?

## ABSTRACT

A central goal of interpretability is to recover representations of causally relevant concepts from the activations of neural networks. The quality of these concept representations is typically evaluated in isolation, and under implicit independence assumptions that may not hold in practice. Thus, it is unclear whether common featurization methods—including sparse autoencoders (SAEs) and sparse probes—recover *disentangled* representations of these concepts. This study proposes a multi-concept evaluation setting where we control the correlations between textual concepts, such as sentiment, domain, and tense, and analyze performance under increasing correlations between them. We first evaluate the extent to which featurizers can learn disentangled representations of each concept under increasing correlational strengths. We observe a one-to-many relationship from concepts to features: features correspond to no more than one concept, but concepts are distributed across many features. Then, we perform steering experiments, measuring whether each concept is independently manipulable. Even when trained on uniform distributions of concepts, SAE features generally affect many concepts when steered, indicating that they are *not* selective nor independent; nonetheless, features affect disjoint subspaces. These results suggest that correlational metrics for measuring disentanglement are generally not sufficient for establishing independence when steering. This underscores the importance of compositional and out-of-distribution evaluations in interpretability research.

## 1    INTRODUCTION

Interpretability centers on understanding how and why neural networks behave how they do. This requires understanding the underlying causal variables and mechanisms that produce observed input–output behaviors; this study centers on causal variable discovery methods. To uncover causal variable representations, it is now common to deploy *featurization methods*, such as sparse autoencoders (SAEs; Olshausen & Field, 1997; Bricken et al., 2023; Huben et al., 2024) and sparse probes (Gurnee et al., 2023). These methods aim to disentangle activation vectors (wherein a dimension can have many meanings) into sparser spaces where there is a more one-to-one relationship between dimensions and concepts.

Most feature extraction studies and benchmarks focus on isolating single concepts or behaviors, such as refusal (Arditi et al., 2024) and truthfulness (Marks & Tegmark, 2024). This tells us whether the concept exists in the model, but it does not tell us to what degree the concept representation is **independent** and **disentangled** from others. How often do feature extractors recover concept representations with high precision? Answers to this question act as a ceiling for our trust in steering methods to induce similar behaviors in novel contexts—i.e., to what degree we have predictive power and control over the model's future behaviors.

This is not a new idea: the fields of causal representation learning (CRL; Schölkopf et al., 2021) and disentangled representation learning (Higgins et al., 2018; Locatello et al., 2019; 2020b) have rich literatures characterizing the assumptions under which it is possible to identify the true latent causal variables for a task. These fields focus on learning a representation from scratch, whereas the goal of interpretability is to derive a simplified causal model of a large and complex neural network

that has already been trained (Geiger et al., 2024). Both lines of work are unified in asking: *in what circumstances is it possible to recover causally efficacious representations?*

Our work builds upon and extends the metrics and evaluation paradigms of CRL to achieve mechanistic interpretability using language models. We use a probabilistic context-free grammar (PCFG) to generate sentences with labeled for multiple concepts. We use this dataset to evaluate empirically successful and popular interpretability methods, including $k$-sparse probes (Gurnee et al., 2023) and sparse autoencoders (Olshausen & Field, 1997; Huben et al., 2024). First, building on CRL, we use correlational evidence to understand to what degree neurons, sparse features, and probes recover **disentangled** representations of ground-truth concepts. To assess their causal behavior, we conduct steering experiments and find that disentangled features do not imply independent manipulability. To characterize this shortcoming, we propose new metrics that evaluate causal criteria we seek from disentangled representations: 1) **independent manipulability**: disentangled features should allow us to steer one and only one concept downstream; 2) **sparse prediction**: features should allow us to accurately predict the

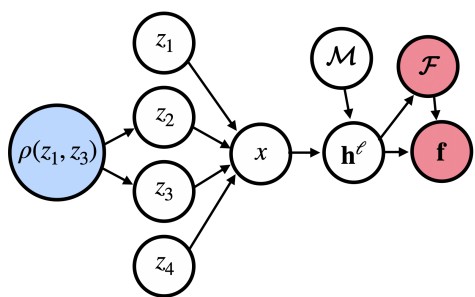

Figure 1: **Causal graph of our experimental setup**. The values of 4 known ground-truth concepts $\{z_i\}_{i=1}^4$ are used to generate an example $x$. We train a featurizer $\mathcal{F}$ to generate vectors $\mathbf{f}$ given activation vectors $\mathbf{h}^\ell$ from the output of layer $\ell$ of language model $\mathcal{M}$. When training $\mathcal{F}$ on examples with increasing correlations between pairs of concepts $\rho(z_i, z_j)$, we observe whether $\mathcal{F}$ learns the true latents or the correlational confound (as measured by the correlation between latents in $\mathbf{f}$ and the presence of the true variable $z_i$).

presence of a concept, ideally with a single feature (Lachapelle et al., 2023a); 3) **disjointness** (Zuheng et al., 2024): steering two concepts jointly should be the sum of steering each concept independently. Our contributions are:

- We design a controlled natural language dataset to evaluate concept disentanglement under confounding. Using this dataset, we demonstrate that commonly-used disentanglement metrics such as MCC (§ 3.1) or DCI-ES (§ 3.2) can characterize the extent to which popular interpretability methods identify the ground-truth concepts, and shed light on when steering is bound to fail.
- We show that disentangled features are *in*sufficient to steer individual features without affecting others and propose new metrics to quantify this shortcoming (§ 4.1)
- We provide causal evidence that existing methods often optimize disjointness, but not independence. That is, they succeed in recovering non-overlapping representations, but often affect multiple unrelated concepts downstream (§ 4.2).

## 2 EXPERIMENTAL SETUP

**Data.** Our goal is to stress-test featurization methods by creating a dataset labeled with known concepts, but where concepts can be correlated to varying degrees. Figure 1 depicts the causal model for our experiments. We vary the correlations between concept-value pairs in the training dataset $\mathcal{D}$ used to train a featurizer $\mathcal{F}$ while holding the language model $\mathcal{M}$ fixed. $\mathcal{F}$ is trained to generate a vector $\mathbf{f}$ of features given activations $\mathbf{h}^\ell$ from layer $\ell$ of language model $\mathcal{M}$. The feature vector $\mathbf{f}$ should ideally encode one concept per dimension.

Using a probabilistic context-free grammar (PCFG), we generate a training dataset $\mathcal{D}$ containing 382,884 sentences and test dataset $\mathcal{T}$ consisting of 1,007 sentences, where each sentence is labeled for 4 concepts $z_i \in \mathcal{Z}$: voice, tense, sentiment, and domain. In our datasets, voice (active, passive) and tense (present, past) are binary. Sentiment (positive, neutral, negative) is multinomial and ordinal, while domain (news, science, fantasy, other) is multinomial with no inherent ordering. Categorical variables will be treated as one-hot vectors of binary values—e.g., $z_i = [v_{i,0}, v_{i,1}, v_{i,2}]$ for sentiment, where $v_{i,0} = 1$ when sentiment is negative and $v_{i,0} = 0$ otherwise.

We fix a target correlation between two concept values—for example, positive sentiment and the science domain—and introduce an unobserved common cause (the blue node in Figure 1) to create

the desired correlation. This creates a confounding variable that acts as the parent of both correlated concepts in the data generating process (DGP). Under varying correlational conditions, we observe to what extent $\mathcal{F}$ can identify the true concepts $\mathcal{Z}$. See App. A for further details on data generation and example sentences.

**Models and featurizers.** A featurizer consists of an encoder $\mathcal{F} : \mathbb{R}^{|\mathbf{h}|} \to \mathbb{R}^{|\mathbf{f}|}$ and optionally a decoder[1] $\mathcal{F}^{-1} : \mathbb{R}^{|\mathbf{f}|} \to \mathbb{R}^{|\mathbf{h}|}$. The encoder $\mathcal{F}$ maps hidden representation vector $\mathbf{h}^{\ell}$ at layer $\ell$ to features $\mathbf{f}$ (where typically, $|\mathbf{f}| > |\mathbf{h}|$). We focus primarily on unsupervised methods such as sparse autoencoders (SAEs), due to their popularity in recent unsupervised interpretability research (Costa et al., 2025; Huben et al., 2024; Mueller et al., 2025a; Marks et al., 2025). We formally define each SAE architecture we test in App. B. To assess how much information about the target concepts is lost relative to a supervised method, we compare to $k$-sparse probes, which are allowed to have non-zero weights to $\leq k$ dimensions of their inputs. Following Gurnee et al. (2023), we first train linear probes with $L_1$ regularization and use the top $k$ weights to find the top $k$ most influential neurons; then, we train logistic regression probes with $L_2$ regularization on those top $k$ neurons.

We focus on two models: Pythia-70M (Biderman et al., 2023) and Gemma-2-2B (Team et al., 2024). We choose these because there exist publicly available SAEs trained on large natural language corpora, including the ReLU SAEs of Marks et al. (2025) and the GemmaScope SAEs (Lieberum et al., 2024).

Recent work has demonstrated the importance of the featurizer's inductive bias, especially when deploying unsupervised featurizers (Hindupur et al., 2025; Costa et al., 2025). We therefore compare SAEs that make varying geometric assumptions: ReLU SAEs (Bricken et al., 2023) assume linear separability, Top-K SAEs (Gao et al., 2025) assume angular separability, and SpADE SAEs (Costa et al., 2025) make weaker assumptions that allow for more heterogeneous concept geometries; we refer readers to App. B for details.

## 3 EVALUATING DISENTANGLEMENT

### 3.1 CONCEPT IDENTIFICATION

A key desideratum of featurizers is the ability to identify the ground-truth concepts despite potential spurious correlations between them.[2] To assess to what degree this property holds for popular featurizers, we design an identifiability evaluation. Intuitively, identifiability measure whether and to what extent the learned model can recover the latent factors that generated the data (e.g., $z_i$ in Figure 1). For formal definitions, see App. C.

**Metrics.** To evaluate the ability of a featurizer to represent these concepts, we employ the **mean correlation coefficient** (MCC) metric (Hyvarinen & Morioka, 2016) common in the causal representation learning literature (Hyvarinen et al., 2019; Khemakhem et al., 2020b;a; Wendong et al., 2023; von Kügelgen et al., 2021; 2023; von Kügelgen, 2024; Reizinger et al., 2024a; 2023b;a; Gresele et al., 2021). MCC measures how well the learned representation recovers the underlying ground-truth factors. That is, it measures identifiability up to scalings and permutations (for details, refer to App. D). One important nuance is that MCC is measured using one-dimensional features, but multinomial concepts may not be one-dimensional in $\mathbf{f}$ or $\mathbf{h}^{\ell}$ (Engels et al., 2025). Thus, we compute the MCC over binarized concepts: for given a variable with $V_i$ possible values, we create a new binary variable for each value of a multinomial concept. For example, for the sentiment concept, we have three binary variables for each of negative, neutral, and positive sentiment. When computing the MCC, we first average the correlation coefficients for all values (e.g., the negative, neutral, and positive binary variables for sentiment) before taking the macroaverage across concepts (e.g., sentiment and tense). A high MCC is achievable in theory only if we make the following assumption:

**Assumption: Linear sufficiency.** *For each ground-truth concept $z_k$, there exists a linear invertible transformation $T$ such that $z_k = T\mathbf{h}^{\ell}$ where $\mathbf{h}^{\ell}$ are the representations of the model $\mathcal{M}$.*

---

[1] Note that this is not a literal inversion. The decoder is typically learned such that the reconstruction error is minimized, but information is lost when reconstructing $\mathbf{h}$ using the featurizer.

[2] We cannot expect a model, supervised or unsupervised, to be able to disentangle two concepts if they are *completely* correlated in the data (Wiedemer et al., 2023) without making any assumptions. However, given at least a couple examples where two concepts do not covary, it is possible in theory to recover independent representations of these concepts.

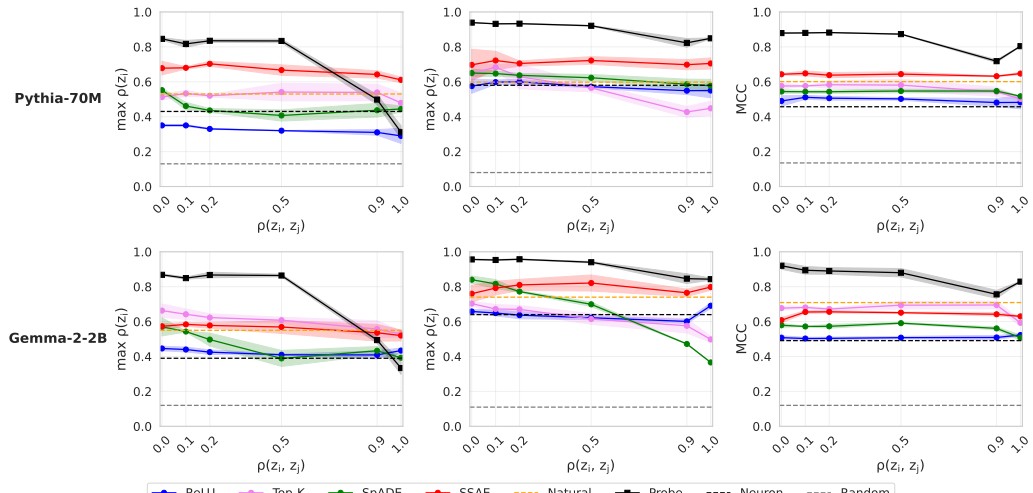

Figure 2: **Maximum correlation coefficient for domain=science (left), sentiment=positive (middle), and MCC (right) under varying correlational conditions**. Shaded regions represent 1 std. dev. across 3 training seeds. Ideal performance looks like a flat line at a high MCC. Probes (supervised featurizers, in black) perform best. SSAEs perform best among unsupervised featurizers. SAEs trained on large-scale natural data (Natural) perform similarly to our best SAEs trained on CFG-generated data, but SSAEs outperform both.

To validate this assumption, we train linear probes for each binary concept and observe whether each probe obtains high accuracy on the concept it was trained to detect, *but also* obtains random-chance accuracy on all other concepts. Our probes satisfy these criteria and thus empirically support Assumption 1; see Figure 9 (App. F).

**Baselines and skylines.** We compare against a randomly initialized SAE (*Random*), the neurons from the residual stream whose correlations correlate most with each concept (*Neuron*, equivalent to the identity featurizer $\mathbf{f} = \mathbf{h}^{\ell}$), and publicly available SAEs trained on natural language data (*Marks* (Marks et al., 2025) and *GemmaScope* (Lieberum et al., 2024) for Pythia-70M and Gemma-2-2B, respectively).

To establish a supervised skyline (*Probe*), we train logistic regression probes using the binarized concept labels. We treat the probe's logit as the feature activation $f_j$ when computing the correlation, and take the average correlation across concept-specific binary probes to compute the MCC.

**Hypothesis.** The ideal result is a high MCC that remains constant as the correlation between ground-truth concepts increases in the training data. We expect unsupervised featurizers, such as SAEs, to perform worse than supervised featurizers. We also expect SAEs trained on our dataset to be better able to isolate the ground-truth concepts compared to the *Natural* baselines; this is because the number of varying concepts is lower, which should make these concepts easier to isolate.

**Results.** Figure 2 shows the MCC for Pythia-70M and Gemma-2-2B for the domain and sentiment concepts as they become more correlated in the training dataset. We find that probes significantly outperform SAEs, as expected. The margin between probes and SAEs is substantial; thus, if one knows *a priori* what concepts one wishes to find, then one should use supervised methods. This agrees with recommendations from Wu et al. (2025) and Mueller et al. (2025b).

SSAEs perform best among unsupervised methods, as hypothesized. Top-K SAEs also perform well among non-contrastive SAE methods with respect to MCC (though they underperform for sentiment=positive). Our SAEs trained on synthetically generated data achieve comparable performance to SAEs trained on a much larger natural language corpus (the *Natural* SAEs in Figure 2); SSAEs outperform them for Pythia-70M, but not for Gemma-2-2B. Most unsupervised methods achieve

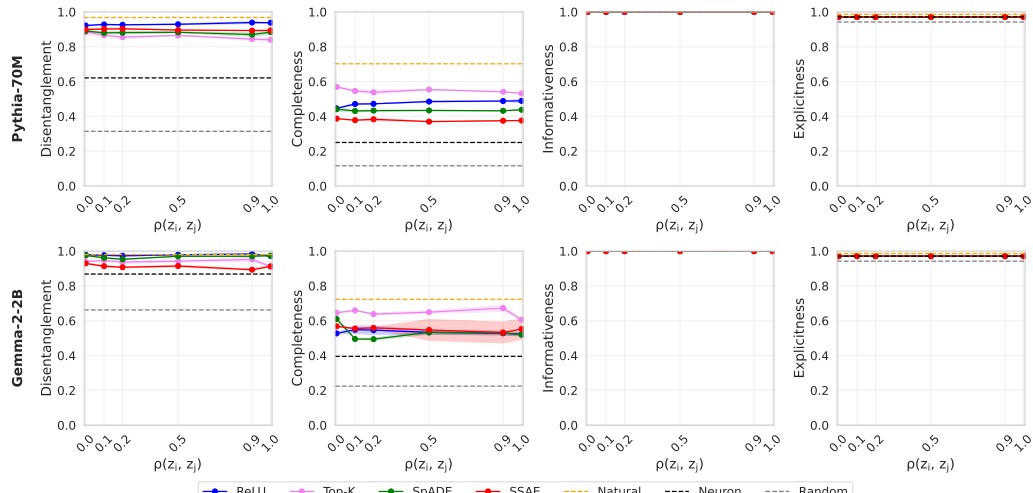

Figure 3: **DCI-ES scores under varying correlational conditions**. Shaded regions represent 1 std. dev. across 3 training seeds. Ideal performance looks like a flat line at high values for all metrics. All methods achieve high disentanglement, informativeness, and explicitness, but relatively low completeness. This suggests that most features capture only one concept, but also that concepts are generally distributed across multiple features (i.e., there is widespread feature splitting).

comparable or lower performance. Thus, for interpreting language models in practice, *one may not need to worry about curating concept-specific data if one's dataset is sufficiently large.*

When do correlations between concepts start to impede concept identification? The answer depends on the method: probes and SpADE (Costa et al., 2025) maintain relatively consistent MCCs up to correlations of 0.5 between concept pairs in the training data. Beyond this, performance begins to degrade. For SSAEs, MCC remains consistent until we reach correlations of 1.0, as its theory predicts (Joshi et al., 2025). In theory, it is always possible to disentangle concepts given at least 2 examples where those concepts do not covary. In practice, however, correlations over 0.5 cause most methods to degrade—including supervised methods. We recommend that future interpretablity studies devote effort to investigating potential correlates of the concept of focus to ensure that other concepts are not being included in learned or derived concept representations.

## 3.2 DIAGNOSING FAILURES IN IDENTIFIABILITY

As we saw in § 3.1, MCC scores are far from ideal in many scenarios. To diagnose the cause of less-than-perfect MCC, we conduct a more fine-grained evaluation.

**Metrics.** We use the DCI-ES framework of Eastwood et al. (2023). **Disentanglement** $D$ measures how many ground-truth concepts $z_j$ are encoded in a single feature $\mathbf{f}_i$; **completeness** $C$ how many feature $\mathbf{f}_i$ are useful to predict a single concept $z_j$; **informativeness** $I$ is inversely proportional to the prediction error of a probe trained on the feature vector; and **explicitness** $E$ captures the trade-off between the probe's capacity and the probe loss. See App. D for detailed definitions.

Most importantly, DCI-ES can indicate whether and to what extent (or equivalence class) identifiability is achieved. Identifiability up to *invertible linear transformations* is achieved if $I = E = 1$; up to permutation and element-wise reparametrization if $D = C = I = 1$; and *up to sign and permutation* if $D = C = I = E = 1$. Importantly, steering is not guaranteed to work when $I = E = 1$, as for steering, we select the single most correlated dimension, which can be a linear mixture of multiple concepts. $D = C = I = E = 1$ implies that all concepts are encoded in a single feature, which means we could predict the impact of steering on concept probabilities via linear extrapolation—even under multiple steering operations.

**Hypothesis.** We hypothesize that all concepts will be recoverable from SAE feature vectors—i.e., that $I$ will be 1 (for more details, see App. D). We also hypothesize that features sensitive to one concept will generally be sensitive to only that concept; this implies that $D$ will be 1. Feature splitting is a known challenge when using SAEs, and we believe it will occur here; thus, we expect $C$ to be significantly less than 1. Because SAEs are trained to be sparse, we expect $E$ to be close to 1.

**Results.** We observe (Figure 3) that $D$, $I$, and $E$ are high for all SAE architectures. This suggests that each SAE identifies the ground-truth concepts up to invertible linear transformation. However, $C$ is low, which suggests that the SAEs do not identify concepts up to sign and permutation. Intuitively, these results imply that all concepts are perfectly recoverable ($I = 1$) with limited expressive power (high $E$). Most features are sensitive to one concept (high $D$), but concepts are often distributed across many features (low $C$).

To what degree does feature splitting occur? To quantify this, we use $k$-sparse probes (Gurnee et al., 2023) and analyze how many features are necessary before probes stop improving. We observe that at least 10 features are needed before returns begin to diminish; see App. E.

High $D, I$, and $E$ suggest that steering these features should only affect the probability of the target concept being steered. Low $C$ suggests that this effect will probably be small, and may not apply to all examples where the concept is present. In the following section, we test this prediction by evaluating whether this happens in a setting where we steer the top SAE features for each concept.

## 4 EVALUATING COUNTERFACTUAL INDEPENDENCE AND DISJOINTNESS

Following the common practice of identifiability evaluations with the MCC only provides correlational evidence and cannot diagnose failure cases (§3.1, whereas the DCI-ES framework (§ 3.2) at least provides more level of detail. However, neither of these correlational measures can provide *causal* evidence that we can independently manipulate concepts using the learned features. Thus, to measure causal efficacy, we employ steering as an independence test of the mechanisms between the features. This can be seen as testing the Independent Causal Mechanism principle prevalent in the causality literature (Pearl, 2009; Peters et al., 2018), which holds that different causal mechanisms neither influence nor inform each other.

### 4.1 STEERING AS A CAUSAL INDEPENDENCE TEST

To locate the steering feature, we could select the feature whose correlation is highest with the label, as in §3.1. However, Arad et al. (2025) has found that the features that detect the input concept (the top correlated features in our case) and the features that control the output concept are distinct. Thus, for steering experiments, we use gradient attributions (Simonyan et al., 2014) to locate the feature that should be steered. We would like features that increase the probability of some concept value $v_{i,x}$; as a proxy, we can fold the featurizer into the forward pass of the model (following Marks et al., 2025), take the logit $\Pi(\mathbf{h}^L)$ of a binary probe $\Pi$ trained on the final layer $L$ of $\mathcal{M}$ to detect a concept value $v_{i,x}$, backpropagate from this logit to obtain its gradient with respect to a feature $\frac{\partial \Pi(\mathbf{h}^L)}{\partial f_i}$, and multiply each feature's gradient by its activation to obtain the gradient attribution $\frac{\partial \Pi(\mathbf{h}^L)}{\partial f_i} \cdot f_i$.[3] We take the feature with the maximum average attribution across examples.

Steering of the activations of layer $\ell$ with the best feature $\hat{\mathbf{f}}_j$ is performed using steering function $\tilde{\mathbf{h}}^\ell(\mathbf{f}_i) \leftarrow \Phi(\mathbf{h}^\ell, \mathcal{F}, i, \alpha)$, where $\Phi$ is defined as follows:

$$\Phi(\mathbf{h}^\ell, \mathcal{F}, i, \alpha) = \mathcal{F}^{-1}\Big(\mathcal{F}(\mathbf{h}^\ell)|\mathrm{do}(\mathbf{f}_i = \alpha \cdot \max(f_i))\Big) + \epsilon \tag{1}$$

where $\alpha$ controls the strength of the steering operation, $\mathcal{F}(\mathbf{h})$ corresponds to the featurized activations, and the do-operation denotes a feature intervention where feature $i$ is set to $\alpha$ times its maximum on

---

[3]Intuitively, this is a first-order Taylor approximation of the effect of changing feature activation $f_i$ to 0 on $\Pi(\mathbf{h}^L)$.

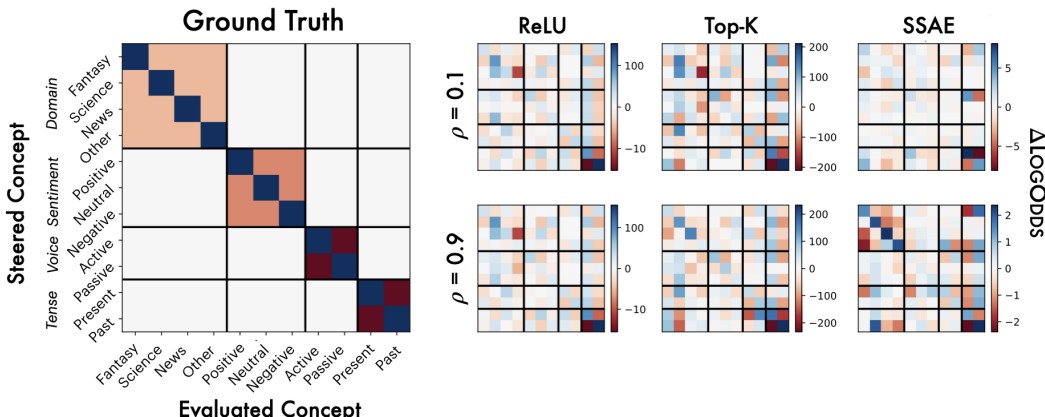

Figure 4: **The effect of steering a given concept (row) on the log-odds of another (column), as measured by a probe**. Results for Pythia-70M shown here; see App. I for Gemma-2-2B. If concept representations are causally independent, we expect a heatmap that resembles the ground-truth: $\Delta$LOGODDS should be high on the diagonal, negative for within-concept pairs, and close to 0.0 for across-concept pairs. All SAEs demonstrate the expected diagonals, but also significant across-concept effects, indicating non-independence. Increasing correlations in the training data, even up to 0.9, do not significantly change the trends.

training dataset $\mathcal{D}$.[4] $\epsilon = \mathbf{h} - \mathcal{F}^{-1}(\mathcal{F}(\mathbf{h}))$ is the reconstruction error without interventions. We set $\alpha$ to 5, but try different values in §4.2.

**Metrics.** For all concept pairs $\{(z_i, z_j) : i, j \in [n]\}$, we steer with $z_i$ and plot $\Delta$LOGODDS of the *other* concept $z_j$.[5] We steer with an SAE trained on the middle layer of $\mathcal{M}$ and then quantify $\Delta$LOGODDS$(z_j)$ as the change in the logit of a multinomial concept probe.[6] We introduce **steering independence** $I_S$ to quantify to what degree a concept is influenced only by its top-attribution feature and no others, whereas **feature selectivity** $S_S$ reflects to what degree a feature only influences its respective concept:

$$I_S = \frac{|\log p(z_i|\tilde{\mathbf{h}}^\ell(\hat{\mathbf{f}}_i)) - \log p(z_i)|}{\sum_j (|\log p(z_i|\tilde{\mathbf{h}}(\hat{\mathbf{f}}_j)) - \log p(z_i)|)}, \qquad S_S = \frac{|\log p(z_i|\tilde{\mathbf{h}}^\ell(\hat{\mathbf{f}}_i)) - \log p(z_i)|}{\sum_j (|\log p((z_j|\tilde{\mathbf{h}}(\hat{\mathbf{f}}_i)) - \log p(z_j)|)}, \qquad (2)$$

$I_S$ and $S_S$ are conceptually similar to normalized pointwise mutual information. In both equations, $j$ excludes within-concept pairs. For example, if $i$ is domain=science, $j$ would skip all other domains.

Mean scores across rows/columns tend to be relatively low, whereas maximal scores are high. High maximal scores indicate that some features can be independently steered with a single feature, but significantly lower means also indicates that many concepts cannot be steered without interference.

**Hypothesis.** If two concepts are independent, then we expect no cross-concept effects—i.e., if two features $\hat{\mathbf{f}}_i$ and $\hat{\mathbf{f}}_{j \neq i}$ correspond to independent concepts $z_i$ and $z_j$, then steering $z_i$ should not change $p(z_j)$. Note that within-concept effects are expected: for $\hat{\mathbf{f}}_i$ and $\hat{\mathbf{f}}_j$ such that $i$ and $j$ are really two values of the same concept $z_i$ (e.g., positive sentiment and negative sentiment), then positive steering with one feature should necessarily decrease the probability of the other. To summarize these heatmaps, we show the **selectivity** and **steering independence** in Table 1.

---

[4]This is equivalent to adding the difference between the steered reconstruction and original reconstruction to the activation.

[5]$\Delta$LOGODDS is equivalent to the logit difference.

[6]These are architecturally similar to the probes used in §3.1, but trained on the *final* layer of $\mathcal{M}$ instead of the middle layer. We use the final layer because it acts as a better proxy for the model's likely output behavior, as opposed to the model's inner representation of the input concepts. We use multinomial probes because they make the change in probabilities for within-concept pairs sum to 1.

**Results.** We observe (Figure 4) that for each SAE architecture, the expected diagonal trend is present, indicating that steering is increasing the log-odds of the target concept as expected. However, in even the best architectures, steering leads to measurable impacts on many unrelated concepts, indicating widespread non-independence. Table 1 quantitatively summarizes these results; best-case scores are high, but mean scores are low, indicating that disentanglement in steering is achieved only for a small subset of concepts.

Table 1: **Steering independence and feature selectivity scores.** We present mean scores per feature/concept, and maxima across features/concepts in parentheses and bold. High independence means that a concept is only influenced by one feature; high selectivity means that a feature only influences one concept. Mean independence and selectivity are generally low, indicating widespread entanglement; however, maximal scores are high, indicating that at least one concept is selectively recovered by these architectures.

| SAE | $\rho$ | Pythia-70M | | Gemma-2-2B | |
|-----|--------|--------------|-------------|--------------|-------------|
|     |        | Independence | Selectivity | Independence | Selectivity |
| ReLU | 0.1 | 0.31 (**0.48**) | 0.30 (**0.89**) | 0.21 (**0.47**) | 0.25 (**1.16**) |
|      | 0.9 | 0.30 (**0.50**) | 0.30 (**0.89**) | 0.23 (**0.62**) | 0.25 (**1.06**) |
| Top-K | 0.1 | 0.29 (**0.76**) | 0.27 (**0.74**) | 0.21 (**0.75**) | 0.25 (**1.21**) |
|       | 0.9 | 0.36 (**1.00**) | 0.30 (**0.61**) | 0.33 (**0.76**) | 0.32 (**1.01**) |
| SSAE | 0.1 | 0.28 (**0.76**) | 0.33 (**0.85**) | 0.22 (**0.29**) | 0.30 (**0.75**) |
|      | 0.9 | 0.42 (**1.41**) | 0.62 (**2.74**) | 0.21 (**0.51**) | 0.25 (**0.98**) |

This underscores the importance of both multi-concept evaluations *and* counterfactual interventions in evaluating concept representations: our correlational analyses did not suggest that interference would be likely in a steering setup, and yet we find evidence of widespread interference. This may align with the findings of Arad et al. (2025): if input features and output features are truly distinct, then identification of the input features may not say anything about our ability to independently steer.

To validate that the concepts can be disentangled in the model, and to validate that probe logits are good proxies for concept presence, we show heatmaps of probe accuracies in Figure 10 (App. F). We observe that each concept probe obtains high performance on its concept's test set, and achieves random-chance performance on all other concepts.

## 4.2 DISJOINTNESS

Steering with one concept and evaluating across many others can provide causal evidence as to how disentangled two concepts are. Now, inspired by Zuheng et al. (2024), we ask whether these concept representations are **disjoint**—i.e., whether they affect non-overlapping subspaces. This is non-equivalent to independence: even if two features correspond to non-overlapping subspaces (i.e, are disjoint), they could still produce non-zero effects on unrelated concepts.

**Metrics.** Disjointness implies that we can predict the effect of pairs of steering operations on $z_i$ from individual steering operations, even if individual steering operations affect multiple concepts. Studying disjointness is important because its presence gives us predictive power over model behavior, even in unseen or potentially out-of-distribution scenarios. See Figure 7 for illustrations and a direct contrast of independence and disjointness. Formally, disjointness is achieved when:

$$p(z_i|\tilde{\mathbf{h}}^\ell(\hat{\mathbf{f}}_i, \hat{\mathbf{f}}_j)) - p(z_i|\mathbf{h}^\ell) = \left(p(z_i|\tilde{\mathbf{h}}^\ell(\hat{\mathbf{f}}_i)) - p(z_i|\mathbf{h}^\ell)\right) + \left(p(z_i|\tilde{\mathbf{h}}^\ell(\hat{\mathbf{f}}_j)) - p(z_i|\mathbf{h}^\ell)\right). \quad (3)$$

That is, the effect on $p(z_i)$ of steering both $\hat{\mathbf{f}}_i$ and $\hat{\mathbf{f}}_j$ should be equivalent to the sum of steering only $\hat{\mathbf{f}}_i$ and $\hat{\mathbf{f}}_j$ in isolation. In practice, we show LOGODDS rather than probabilities; this unbounded metric is more likely to be additive at especially high and low probabilities due to greater numeric precision.

**Hypothesis.** Under low correlations, we expect that concepts will be disjoint, such that the effect of steering the top features for $z_i$ and $z_j$ on $\Delta$LOGODDS($z_i$) will be additive, regardless of their (non-)independence. Under higher correlations, we expect less disjoint representations and more non-linearly predictable interaction terms between pairs of steering operations.

**Results.** We observe (Figure 5) that the effect of steering with two concepts simultaneously is almost exactly equivalent to summing the impact of steering with both concepts separately. To quantitatively verify, we compute the $R^2$ for each SAE; this measures how This suggests no interaction terms.

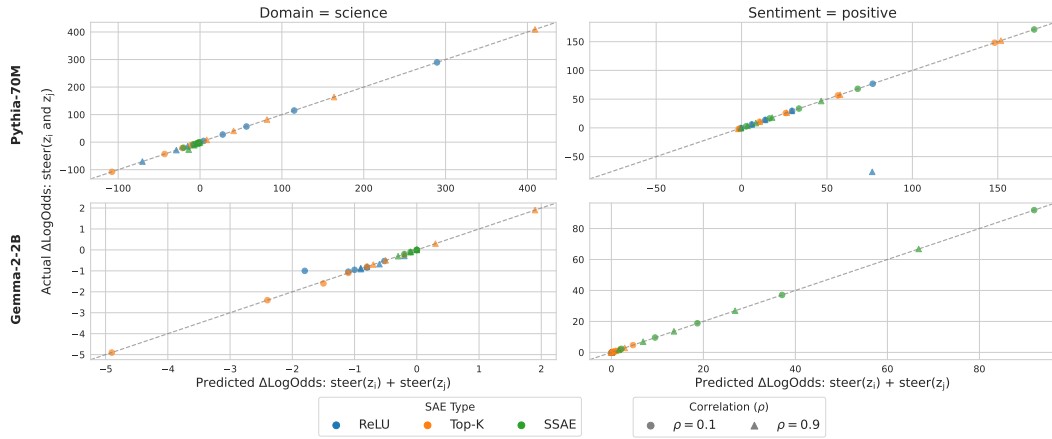

Figure 5: **Predicted $\Delta\text{LogOdds}(z_i)$ under disjointness assumptions vs. actual $\Delta\text{LogOdds}(z_i)$ when steering relevant feature $\hat{\mathbf{f}}_i$ and unrelated feature $\hat{\mathbf{f}}_j$.** Predictions are obtained by adding the $\Delta\text{LogOdds}(z_i)$ when steering with either $\hat{\mathbf{f}}_i$ or $\hat{\mathbf{f}}_j$ separately; actual values are obtained by steering with both simultaneously. $\hat{\mathbf{f}}_i$ and $\hat{\mathbf{f}}_j$ are typically disjoint, as indicated by the predicted change almost perfectly matching the true change. Disjointness does *not* imply independence; see App. I.

This in combination with the non-independence results of §4.1 suggests that each SAE feature operates on a separate subspace, but also that steering with a feature can still affect representations of other concepts. See Figure 14 (App. I) for a more direct empirical demonstration that disjointness does not imply independence.

Table 2: $R^2$ **between predicted and actual $\Delta\text{LogOdds}(z_i)$ for each SAE.** Values are all near 1.00, indicating near-perfect disjointness for each SAE, even under relatively high correlations between concepts.

| SAE | $\rho$ | Domain=sentiment | | Sentiment=positive | |
| | | Pythia-70M | Gemma-2-2B | Pythia-70M | Gemma-2-2B |
|---|---|---|---|---|---|
| ReLU | 0.1 | 1.00 | 0.93 | 1.00 | 1.00 |
| | 0.9 | 1.00 | 0.94 | 1.00 | 1.00 |
| Top-K | 0.1 | 1.00 | 0.99 | 1.00 | 0.98 |
| | 0.9 | 1.00 | 1.00 | 1.00 | 0.99 |
| SSAE | 0.1 | 1.00 | 1.00 | 1.00 | 1.00 |
| | 0.9 | 1.00 | 1.00 | 1.00 | 1.00 |

## 5 RELATED WORK

**Featurization in interpretability.** In interpretability, *featurization* refers to techniques that allow one to map from less interpretable and denser model representations—typically *neurons*—to more interpretable (and often sparser) representations—what are often called *features*. This has produced supervised techniques such as sparse probing (Gurnee et al., 2023), unsupervised techniques such as sparse autoencoders (SAEs; Olshausen & Field, 1997; Bricken et al., 2023; Huben et al., 2024), and non-parametric techniques such as steering vectors (Subramani et al., 2022) derived via difference-in-means (Marks & Tegmark, 2024).

How can one evaluate the quality of a feature? Recent work has proposed standardized evaluations based on known concepts (Mueller et al., 2025b; Huang et al., 2024). These allow one to assess whether a concept discovery method discovers a concept with high recall. However, it leaves precision unexplored: how well do these concept representations disentangle the concept from others? Evaluating this requires multi-concept evaluations, as we propose.

**Causal representation learning.** Causal representation learning (CRL; Schölkopf et al., 2021) assumes that high-dimensional observations, such as text, are generated from low-dimensional latent factors, whose relationships to other latent factors are encoded in a causal graph. Then, CRL proposes latent variable models of such observations that are **identifiable**, meaning that the recovered features (and possibly a graph over them) are related to the true factors up to permutation and element-wise transformations. Since such unsupervised learning is not identifiable without further assumptions (Hyvärinen & Pajunen, 1999; Darmois, 1951; Locatello et al., 2019), CRL methods rely on non-iid data or constraints on the decoding function (Moran et al., 2022; Gresele et al., 2021; Lachapelle et al., 2023b; Brady et al., 2025; Reizinger et al., 2023b). For example, CRL has developed identifiable

models using data from sparse interventions Ahuja et al. (2022b); Zhang et al. (2023); Buchholz et al. (2023); von Kügelgen et al. (2023), contrastive pairs of samples (Ahuja et al., 2022a; Locatello et al., 2020a; Gresele et al., 2019; Brehmer et al., 2022), data from multiple environments (Ahuja et al., 2023; Layne et al., 2025; Khemakhem et al., 2020a), and temporal data with sparse or intervened mechanisms (Lachapelle et al., 2021; Lippe et al., 2023; 2022). We go further, however, and test the causal implications of disentangled features in model outputs: target concept steering, accuracy with sparse probes and disjoint steering effects. Similar to what we propose, Joshi et al. (2025) propose a method that enables identifiable steering under multi-concept shifts; this method performs best on disentanglement *and* steering-based metrics.

To corroborate the theoretical claims of identifiability, access to the ground-truth factors is required, which generally limits the tasks that can be considered. Among the evaluation metrics, the MCC score (Hyvarinen & Morioka, 2016) has been used widely, despite its shortcomings [cite IEEE, InfoMEC]. Several other metrics have been proposed in both the disentanglement and the identifiable (causal) representation learning communities, such as the IRS score that measures interventional effects (Suter et al., 2019), or the DCI (Eastwood & Williams, 2018), DCI-ES (Eastwood et al., 2023), and InfoMEC (Hsu et al., 2023) scores that directly aim to improve on the MCC.

**Compositional generalization.** Closely related to disentanglement and the notion of disjoint effects is the ability of models to compose concepts in novel ways, called compositional generalization. Compositional generalization has a long history in the NLP literature (Ahuja & Mansouri, 2024; Han & Padó, 2024; Ramesh et al., 2024; Lake & Baroni, 2023; Nogueira et al., 2021; Dziri et al., 2023; Saparov et al., 2023; Mészáros et al., 2024; Reizinger et al., 2024b; Ujváry et al., 2025), but tends to focus on the reuse of syntactic chunks or lexemes. Some recent CRL studies investigate compositional latents; they tend to study simplified formal languages, such as regular languages or Dyck (bracketing) languages (Deletang et al., 2022; Mészáros et al., 2024; Reizinger et al., 2024b; Ujváry et al., 2025).

# 6 DISCUSSION AND CONCLUSIONS

Each of our experiments has revealed insufficiencies in single-concept evaluations. One may achieve far above random-chance performance under correlational evaluation methods (§3.1) and improvements in sparsity over the native residual representation space a model (§E). Even so, causal evidence reveals that entanglement can still be likely and widespread (§4.1,4.2) even when the aforementioned correlational metrics suggest otherwise.

Despite strong entanglement, concept pairs demonstrated very little interaction effects (§4.2). This implies that when features achieve the *form* of separation—that is, that the cosine similarity of the subspace on which they act is very low—it does not necessarily imply that their *functional roles* are non-interacting. This suggests that mechanistic interpretability studies aiming to establish the independence of two mechanisms cannot settle for establishing that subspaces or circuits do not overlap; one must directly establish that the functional roles on the final output are independent.

One dimension is not sufficient, even with methods with strong sparsity regularizers. This may imply that the intrinsic dimensionalities of the concepts themselves are greater than one. Given the variance of scientific domains or positive sentiment, this would not necessarily be surprising. It would be interesting for future work to investigate the relationship between causal independence metrics and the intrinsic dimensionality of feature representations—for example, using techniques like those of Engels et al. (2025). Broadly speaking, more work is needed on methods for detecting, characterizing, and steering with multi-dimensional concepts.

**Limitations.** We acknowledge that this study could be improved in multiple respects. Our data is generated by a CFG; while it is natural language, it could still be OOD with respect to the data one would normally find in a corpus. We also only study categorical concepts; while reasonable to keep the study tractable, we believe that continuous concepts could yield interesting results.

## REPRODUCIBILITY

To ensure the robustness of our results, we average results across three random seeds and report standard deviations. For all optimization-based procedures, we fix and save these random seeds; these settings will be released alongside our code. We will release all code and data upon deanonymization.

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

```
S[active, present] -> Subj V O | Subj V O PP | Adv, Subj V O

S[passive, past] -> Subj V_past O
                  | Subj V_past O PP
                  | Adv, Subj V_past O

S[passive, present] -> O is V_pp by Subj
                     | Adv, O is V_pp
                     | O is being V_pp by Subj

S[passive, past] -> O was V_pp by Subj
                  | Adv, O was V_pp
                  | O had been V_pp by Subj

Subj[news, positive] -> the successful team
                      | the innovative company

Subj[news, neutral] -> the government | the company

Subj[fantasy, negative] -> the evil sorcerer
                         | the treacherous assassin

V[negative] -> criticizes | condemns | rejects

V[neutral] -> announces | reports | explains

V_past[positive] -> celebrated | praised | endorsed

PP[neutral] -> in recent days
             | across different sectors

PP[positive] -> with remarkable success
              | beyond expectations

PP[negative] -> without proper justification
              | to widespread criticism
```

Figure 6: Excerpts from the context-free grammar we use to generate our SAE training and evaluation datasets.

## A  DATA GENERATION

We use probabilistic context-free grammars to generate the training data for our SAEs. Non-terminals have attributes corresponding to the ground-truth concepts. In Figure 6, we show a subsample of the rules in the grammar. Note that this sample is simplified: most terminal-generating rules have over 10 non-terminals, and there are more sentence templates than displayed in the figure.

In Table 3, we show examples from our generated training set. When we generate without correlations between concepts, there is an approximately uniform distribution of each concept, and correlations of approximately 0 across all concept pairs. If a concept-value pair is correlated, we pre-compute the example set such that we can achieve the closest match to the desired correlation. When training SAEs, we iterate for multiple epochs over the full dataset (when there are no cross-concept correlations) or the subsampled dataset (when there are cross-concept correlations).

## B  SAE TRAINING DETAILS

### B.1  SAE ARCHITECTURES

Here, we define sparse autoencoders and describe the differences between the architectures we study.

Table 3: Examples of sentences generated by our context-free grammar.

| Concept Label | | | | Example Sentence |
|---|---|---|---|---|
| Voice | Tense | Domain | Sentiment | |
| Active | Present | Science | Positive | The brilliant scientist celebrates the remarkable findings. |
| Active | Present | Science | Neutral | The expert announces the parameters in recent days. |
| Active | Present | Science | Negative | As of today, the discredited theory rejects the inconclusive evidence. |
| Active | Past | Fantasy | Negative | Unsuccessfully, the malevolent dragon damaged the corrupted land. |
| Passive | Past | News | Neutral | The event was explained in the recent report. |
| Passive | Present | Other | Positive | The pleasant surprise is endorsed advantageously by the talented artist. |
| Passive | Past | Other | Neutral | The question was answered when the family announced the event. |

**Sparse autoencoders.** The conceptually simplest architecture we deploy is the ReLU sparse autoencoder (Huben et al., 2024; Bricken et al., 2023), which learns a mapping from $\mathbf{x} = \mathbf{h}^\ell$ to a learned sparse feature vector $\mathbf{f}$, and then reconstructs the activations $\hat{\mathbf{x}}$ given $\mathbf{f}$. More formally:

$$\mathbf{f} = \text{ReLU}(W_{\text{enc}}\mathbf{x} + \mathbf{b}_{\text{enc}}) \tag{4}$$

$$\hat{\mathbf{x}} = W_{\text{dec}}(\mathbf{f} - \mathbf{b}_{\text{enc}}) + \mathbf{b}_{\text{dec}} \tag{5}$$

ReLU SAEs minimize $\mathcal{L} = \text{MSE}(\mathbf{x}, \hat{\mathbf{x}}) + \lambda\|\mathbf{f}\|_1$.

Top-K SAEs (Gao et al., 2025) are similar to ReLU SAEs, but they strictly retain the top $k$ activations per sample and zero out all others:

$$\mathbf{f} = \text{top-}k(W_{\text{enc}}\mathbf{x} + \mathbf{b}_{\text{enc}}) \tag{6}$$

Sparsemax distance encoders (SpADE) can capture nonlinearly separable and heterogeneous features; we refer readers to Hindupur et al. (2025) for details. In formal terms:

$$\mathbf{f} = \text{Sparsemax}(-\lambda d(\mathbf{x}, W)) \tag{7}$$

where $d(\mathbf{x}, W)_i = \|\mathbf{x} - W_i\|_2^2$. Hindupur et al. (2025) show that this architecture can capture more irregular concept geometries, whereas ReLU SAEs assume linear separability, and Top-K SAEs assume angular separability.

**Sparse shift autoencoders.** Sparse shift autoencoders (SSAEs) (Joshi et al., 2025) are trained using paired observations $(\boldsymbol{x}, \tilde{\boldsymbol{x}})$ assumed to be sampled from the following generative process:

$$S \sim p(S), \quad (\boldsymbol{c}, \tilde{\boldsymbol{c}}) \sim p(\boldsymbol{c}, \tilde{\boldsymbol{c}} \mid S), \tag{8}$$

$$\boldsymbol{x} \coloneqq g(\boldsymbol{c}), \quad \tilde{\boldsymbol{x}} \coloneqq g(\tilde{\boldsymbol{c}}), \tag{9}$$

where $S \subseteq \{1, \dots, d_c\}$ denotes the subset of concepts that vary between $\boldsymbol{x}$ and $\tilde{\boldsymbol{x}}$ and $d_c$ represents the dimension of *varying concepts*, the concepts that are intervened upon in the dataset.

Note that SSAEs take as input *difference vectors* $\Delta\mathbf{z} \coloneqq f(\tilde{\boldsymbol{x}}) - f(\boldsymbol{x}) = \tilde{\boldsymbol{z}} - \boldsymbol{z}$ that represent concept differences in activation space and model them as:

$$\Delta\hat{\mathbf{c}}_V \coloneqq r(\Delta\mathbf{z}) \coloneqq \mathbf{W}_e(\Delta\mathbf{z} - \mathbf{b}_d) + \mathbf{b}_e ; \tag{10}$$

$$\Delta\hat{\mathbf{z}} \coloneqq q(\Delta\hat{\mathbf{c}}_V) \coloneqq \mathbf{W}_d\Delta\hat{\mathbf{c}}_V + \mathbf{b}_d \tag{11}$$

where $r : \mathbb{R}^{d_z} \to \mathbb{R}^{d_c}$ is an affine encoder $q : \mathbb{R}^{d_c} \to \mathbb{R}^{d_z}$ is an affine decoder. In words, the representation $r(\Delta\mathbf{z})$ predicts $\Delta\mathbf{c}_V$, i.e., the concept shifts corresponding to $\Delta\mathbf{z}$.

SSAEs are trained to solve the following constrained problem:

$$(\hat{r}, \hat{q}) \in \arg\min_{r,q} \mathbb{E}_{\boldsymbol{x}, \tilde{\boldsymbol{x}}} \left[ ||\Delta\mathbf{z} - q(r(\Delta\mathbf{z}))||_2^2 \right] \tag{12}$$

$$\text{s.t. } \mathbb{E}_{\boldsymbol{x}, \tilde{\boldsymbol{x}}} ||r(\Delta\mathbf{z})||_0 \leq \beta , \tag{13}$$

where Eq. 12 is the standard auto-encoding loss that encourages good reconstruction and Eq. 13 is a regularizer that encourages the predicted concept shift vector $\Delta\hat{\mathbf{c}}_V := \hat{r}(\Delta\mathbf{z})$ to be sparse. Since the $\ell_0$-norm is non-differentiable, in practice we replace it by an $\ell_1$-norm leading to the following relaxed sparsity constraint:

$$\mathbb{E}_{\boldsymbol{x},\tilde{\boldsymbol{x}}}||r(\Delta\mathbf{z})||_1 \leq \beta. \tag{14}$$

We then approximately solve this constrained problem by finding a saddle point of its Lagrangian using the ExtraAdam algorithm (Gidel et al., 2020) as implemented by Gallego-Posada & Ramirez (2022).

### B.2 HYPERPARAMETERS

**Sparse autoencoders.** All Pythia-70M sparse autoencoders are trained using a batch size of 128 sequences for 10000 steps. We train on the output of the middle layer (layer 3). Optimization is performed using Adam with an initial learning rate of $1 \times 10^{-3}$, 200 warmup steps, and $\beta_1 = 0.9, \beta_2 = 0.95$. Top-$k$ SAEs are trained with $k = 128$. For Gemma-2-2B, we use the same hyperparameters for all SAEs except SpADE, which has higher memory requirements; for this architecture, we reduce the batch size to 64 while maintaining all other hyperparameters.[7] We also train on the middle layer (layer 13). Our implementation is based on that of Hindupur et al. (2025).

**Sparse shift autoencoders.** For SSAEs, we must train on pairwise differences in activations. For this, we iterate over the training set to get example $x_i$, and then uniformly sample another example $x_j$, ensuring that $i \neq j$. Otherwise, we use similar hyperparameters as when training SAEs. Note that SSAEs should be trained on the *final* layer of a model, rather than the middle layer: this choice is motivated by the claim that concepts in the output space are most easily linearly identified in the final layer (Joshi et al., 2025).[8]

We present NMSE, variance explained, and percent sparsity on the test set in Table 4 (for Pythia) and Table 5 (for Gemma).

**Probes.** All probes are logistic regression probes. The probes used in correlational experiments are trained on the middle layer of Pythia-70M or Gemma-2-2B for a maximum of 1000 steps. We use the implementation of `scikit-learn` (Pedregosa et al., 2011).[9] $k$-sparse probes are identical in architecture and hyperparameters, but we filter the set of neurons or features to reduce dimensionality before training the probes (and also train them on featurized representations rather than the original activation space); see App. G for details. For the binary probes, we balance the training dataset of each probe by uniformly subsampling the more frequent class such that the number of examples for both classes is the same.

For the multinomial probes used for evaluating steering, the architecture and hyperparameters are the same, except that the probe outputs one logit *per concept value* rather than a single logit. These probes are trained on the final layer of Pythia-70M or Gemma-2-2B, as their purpose is to estimate the probability of a concept appearing in the model's output. Note that we do not rebalance the data for multinomial probes; we only train multinomial probes on data where there are no cross-concept correlations, so there is already an approximately uniform distribution of labels for each probe's training set.

## C IDENTIFIABILITY DEFINITIONS

Identifiability definitions formulate the permissible transformations—termed an equivalence class—of the learned latent factors **f** by such that the resulting probability distributions parametrized by

---

[7]We experimented with doubling the number of training steps to compensate for the halved batch size for Gemma-2-2B SpADE SAEs. Final loss reductions were very small, so we chose to continue using 10000 iterations for uniformity.

[8]We acknowledge that using different layers for different SAE architectures introduces a confound. However, in pilot experiments, we found that other architectures tended to yield worse disentanglement and steering results when trained on the final layer. Thus, the current locations seem to be closer to optimal than if we had used the same location.

[9]Specifically, we use the Newton-Cholesky solver.

Table 4: Variance explained, losses, and sparsities for SAEs trained on the middle layer of Pythia-70M (or last layer in the case of SSAEs). SSAE results are not comparable to those of other SAEs; unlike other architectures, they are trained and evaluated on *pairwise differences* of activations.

| SAE Arch. | $\rho(z_i, z_j)$ | NMSE | Var. Explained | % Sparsity |
|---|---|---|---|---|
| ReLU | 0.0 | 0.004 (0.000) | 99.6 (0.0) | 58.7 (0.2) |
| | 0.1 | 0.006 (0.000) | 99.7 (0.0) | 54.6 (0.1) |
| | 0.2 | 0.006 (0.008) | 99.7 (0.0) | 54.4 (0.2) |
| | 0.5 | 0.007 (0.000) | 99.7 (0.0) | 54.4 (0.1) |
| | 0.9 | 0.003 (0.000) | 99.7 (0.0) | 54.4 (0.1) |
| | 1.0 | 0.003 (0.000) | 99.7 (0.1) | 54.4 (0.1) |
| Top-K | 0.0 | 0.058 (0.000) | 94.2 (0.0) | 97.6 (0.0) |
| | 0.1 | 0.056 (0.000) | 94.4 (0.0) | 97.6 (0.0) |
| | 0.2 | 0.056 (0.000) | 94.4 (0.0) | 97.6 (0.0) |
| | 0.5 | 0.060 (0.000) | 94.0 (0.0) | 97.6 (0.0) |
| | 0.9 | 0.057 (0.000) | 94.4 (0.0) | 97.6 (0.0) |
| | 1.0 | 0.064 (0.000) | 93.6 (0.0) | 97.6 (0.0) |
| SpADE | 0.0 | 0.003 (0.000) | 99.7 (0.0) | 58.8 (0.3) |
| | 0.1 | 0.003 (0.000) | 99.7 (0.0) | 58.8 (0.2) |
| | 0.2 | 0.003 (0.000) | 99.7 (0.0) | 59.3 (0.1) |
| | 0.5 | 0.003 (0.000) | 99.7 (0.0) | 60.6 (0.0) |
| | 0.9 | 0.004 (0.000) | 99.6 (0.0) | 64.9 (0.1) |
| | 1.0 | 0.005 (0.000) | 99.5 (0.0) | 66.8 (0.1) |
| Natural | 0.005 | 99.5 | 99.8 | |
| SSAE | 0.1 | 0.004 (0.001) | 99.6 | 98.8 (0.0) |
| | 0.1 | 0.004 (0.001) | 99.6 (0.0) | 99.1 (0.0) |
| | 0.2 | 0.005 (0.001) | 99.6 (0.0) | 99.0 (0.0) |
| | 0.5 | 0.005 (0.001) | 99.6 (0.0) | 99.1 (0.0) |
| | 0.9 | 0.005 (0.002) | 99.6 (0.0) | 99.0 (0.0) |
| | 1.0 | 0.004 (0.001) | 99.6 (0.0) | 99.2 (0.0) |

Table 5: Variance explained, losses, and sparsities for SAEs trained on the middle layer of Gemma-2-2B (or last layer in the case of SSAEs). SSAE results are not comparable to those of other SAEs; unlike other architectures, they are trained and evaluated on *pairwise differences* of activations.

| SAE Arch. | $\rho(z_i, z_j)$ | Var. Explained | NMSE | % Sparsity |
|---|---|---|---|---|
| ReLU | 0.0 | 0.014 (0.000) | 98.7 (0.0) | 49.6 (0.1) |
| | 0.1 | 0.014 (0.000) | 98.6 (0.0) | 49.6 (0.1) |
| | 0.2 | 0.014 (0.000) | 98.7 (0.0) | 49.6 (0.0) |
| | 0.5 | 0.014 (0.000) | 98.7 (0.0) | 49.9 (0.0) |
| | 0.9 | 0.011 (0.000) | 98.9 (0.0) | 50.0 (0.0) |
| | 1.0 | 0.010 (0.000) | 99.0 (0.0) | 50.0 (0.0) |
| Top-K | 0.0 | 0.218 (0.001) | 78.1 (0.001) | 99.4 (0.0) |
| | 0.1 | 0.218 (0.000) | 78.1 (0.000) | 99.4 (0.0) |
| | 0.2 | 0.216 (0.001) | 78.3 (0.000) | 99.4 (0.0) |
| | 0.5 | 0.218 (0.000) | 78.2 (0.000) | 99.4 (0.0) |
| | 0.9 | 0.236 (0.000) | 76.4 (0.000) | 99.4 (0.0) |
| | 1.0 | 0.269 (0.000) | 73.1 (0.000) | 99.4 (0.0) |
| SpADE | 0.0 | 0.094 (0.0) | 90.6 (0.0) | 96.9 (0.1) |
| | 0.1 | 0.091 (0.000) | 90.5 (0.0) | 96.8 (0.0) |
| | 0.2 | 0.091 (0.001) | 90.4 (0.0) | 96.9 (0.0) |
| | 0.5 | 0.099 (0.001) | 89.5 (0.0) | 96.7 (0.0) |
| | 0.9 | 0.149 (0.000) | 84.4 (0.1) | 96.9 (0.0) |
| | 1.0 | 0.167 (0.001) | 84.5 (0.1) | 96.2 (0.0) |
| Natural | 0.064 | 93.6 | 99.6 | |
| SSAE | 0.0 | 0.064 (0.001) | 98.8 (0.0) | 91.9 (0.1) |
| | 0.1 | 0.068 (0.000) | 98.8 (0.0) | 91.5 (0.0) |
| | 0.2 | 0.061 (0.000) | 98.8 (0.0) | 91.6 (0.1) |
| | 0.5 | 0.072 (0.000) | 98.8 (0.0) | 91.4 (0.0) |
| | 0.9 | 0.069 (0.000) | 98.9 (0.0) | 91.3 (0.0) |
| | 1.0 | 0.074 (0.001) | 99.0 (0.0) | 91.4 (0.0) |

the neural network are equivalent. The smaller the equivalence class, the stronger assumptions are generally required.

**Definition 1** (Strong Identifiability (Khemakhem et al., 2020b)). *Given a parameter class $\Theta$, when the feature extractors $\mathcal{F}_{\theta_1}, \mathcal{F}_{\theta_2}$ produce latent representations $\mathbf{f}_1 = \mathcal{F}_{\theta_1}(\mathbf{x}), \mathbf{f}_2 = \mathcal{F}_{\theta_2}(\mathbf{x})$ from observations $\mathbf{x}$ that are equivalent up to scaled permutations and offsets $c$ for all $\theta_1, \theta_2 \in \Theta$, i.e.,*

$$\theta_1 \sim \theta_2 \iff \mathbf{f} = \mathcal{F}_{\theta_1}(\mathbf{x}) = \mathbf{DP}\mathcal{F}_{\theta_2}(\mathbf{x}) + c, \tag{15}$$

*where $\mathbf{D}$ is a diagonal and $\mathbf{P}$ a permutation matrix. Then $\theta_1, \theta_2$ fulfill an equivalence relationship.*

**Definition 2** (Weak Identifiability (Khemakhem et al., 2020b)). *Given a parameter class $\Theta$, when the feature extractors $\mathcal{F}_{\theta_1}, \mathcal{F}_{\theta_2}$ produce latent representations $\mathbf{f}_1 = \mathcal{F}_{\theta_1}(\mathbf{x}), \mathbf{f}_2 = \mathcal{F}_{\theta_2}(\mathbf{x})$ from observations $\mathbf{x}$ that are equivalent up to matrix multiplications and offsets $c$ for all $\theta_1, \theta_2 \in \Theta$, i.e.,*

$$\theta_1 \sim \theta_2 \iff \mathbf{f} = \mathcal{F}_{\theta_1}(\mathbf{x}) = \mathbf{A}\mathcal{F}_{\theta_2}(\mathbf{x}) + c, \tag{16}$$

*where $\mathrm{rank}(\mathbf{A}) \geq \min(\dim \mathbf{f}; \dim \mathcal{X})$. Then $\theta_1, \theta_2$ fulfill an equivalence relationship.*

**Definition 3** (Identifiability up to elementwise nonlinearities (Hyvarinen & Morioka, 2017)). *Given a parameter class $\Theta$, when the feature extractors $\mathcal{F}_{\theta_1}, \mathcal{F}_{\theta_2}$ produce latent representations $\mathbf{f}_1 = \mathcal{F}_{\theta_1}(\mathbf{x}), \mathbf{f}_2 = \mathcal{F}_{\theta_2}(\mathbf{x})$ from observations $\mathbf{x}$ that are equivalent up to elementwise nonlinearities, matrix multiplications and offsets $c$ for all $\theta_1, \theta_2 \in \Theta$, i.e.,*

$$\theta_1 \sim \theta_2 \iff \mathbf{f} = \mathcal{F}_{\theta_1}(\mathbf{x}) = \mathbf{A}\sigma[\mathcal{F}_{\theta_2}(\mathbf{x})] + c, \tag{17}$$

*where $\mathrm{rank}(\mathbf{A}) \geq \min(\dim \mathbf{f}; \dim \mathcal{X})$ and $\sigma$ denotes an elementwise nonlinear transformation. Then $\theta_1, \theta_2$ fulfill an equivalence relationship.*

# D METRICS

## D.1 MCC

The Mean Correlation Coefficient (MCC) (Hyvarinen & Morioka, 2016) is a widely used metric to measure how well the learned representation recovers the underlying ground-truth factors. That is, it measures identifiability up to scalings and permutations.

Given a set of ground-truth concepts $\{z_1, \ldots, z_n\}$ that generate an input example $\mathbf{x}$ where each concept $z_j \in \mathbb{Z}$, then $\forall i \in [1, \ldots, n]$, we compute $\hat{\mathbf{f}}_j = \arg\max_i |\rho_{\mathcal{D}}(f_i, z_j)|$, where $f_i$ is the activation of feature $\mathbf{f}_i$ and $\rho$ is the correlation. Intuitively, $\hat{\mathbf{f}}_j$ is the feature whose activation correlates most with the value of $z_j$ on some training dataset $\mathcal{D}$. Given test set $\mathcal{T}$ where concepts are uniformly distributed w.r.t. each other (i.e., no built-in correlations), we use $\rho_{\mathcal{T}}(\hat{f}_j, z_j)$ as a measure of how well the featurizer linearly identifies concept $z_j$. After locating the best features $\{\hat{\mathbf{f}}_j\}_{j=1}^n$ for each concept, we compute the MCC as the mean of their correlations with their respective concepts on $\mathcal{T}$. In other words:

$$\mathrm{MCC} = \frac{1}{n}\sum_{j=1}^n \rho_{\mathcal{T}}(\hat{f}_j, z_j). \tag{18}$$

The MCC is measured using one-dimensional features, but multinomial concepts may not be one-dimensional in $\mathbf{f}$ or $\mathbf{h}^\ell$ (Engels et al., 2025). Thus, to create a fairer evaluation, we compute the MCC over binarized concepts. That is, given a variable $z_i \in \mathbb{Z}$ with $V_i$ possible values, we create a new binary variable $v_{i,x} \in \mathbb{B}$ for each value $x$ corresponding to whether $z_i = v_{i,x}$. When computing the MCC, we first average the correlation coefficients for all $v_{i,x} \in V_i$ before taking the macroaverage across concepts.

## D.2 DCI-ES

Here, we summarize the DCI-ES metrics (Eastwood et al., 2023), and give methodological details as to how we compute them. Our implementation is based directly on that of Eastwood et al. (2023).

DCI-ES stands for **d**isentanglement, **c**ompleteness, **i**nformativeness, **e**xplicitness, and **s**ize. We focus on the first four metrics, as these are the most relevant to establishing identifiability. Disentanglement

$$p(z_i|\mathbf{h}^\ell) \xrightarrow{\Phi(\mathbf{h}^\ell,\mathcal{F},i,\alpha)} p(z_i|\tilde{\mathbf{h}}^\ell(\hat{\mathbf{f}}_i))$$

$$\Phi(\mathbf{h}^\ell,\mathcal{F},j,\beta)\Big\downarrow \qquad\qquad \Big\downarrow\Phi(\mathbf{h}^\ell,\mathcal{F},j,\beta) \qquad\qquad p(z_j|\mathbf{h}^\ell)\underset{\text{no change}}{\overset{\Phi(\mathbf{h}^\ell,\,\mathcal{F},\,i,\,\alpha)}{\xrightarrow{\hspace{2cm}}}}p(z_j|\tilde{\mathbf{h}}^\ell(\hat{\mathbf{f}}_i))$$

$$p(z_i|\tilde{\mathbf{h}}^\ell(\hat{\mathbf{f}}_j)) \xrightarrow[\Phi(\mathbf{h}^\ell,\mathcal{F},i,\alpha)]{} p(z_i|\tilde{\mathbf{h}}^\ell(\hat{\mathbf{f}}_i,\hat{\mathbf{f}}_j))$$

Figure 7: **The difference between feature disjointness and independence: (Left)** Two concepts $z_i$ and $z_j$ with feature representations $\hat{\mathbf{f}}_i$ and $\hat{\mathbf{f}}_j$, respectively, are disjoint if the left diagram commutes. **(Right)** If they are independent then there is no commutative relationship, as steering with $\hat{\mathbf{f}}_i$ should not affect $p(z_j)$. Intuitively, disjointness implies that two feature representations exist in non-overlapping subspaces of the model representations, and thus that the effect of steering of both can be predicted from the result of steering either in isolation. Independence implies that steering with one concept would not affect how the model uses other concepts. Refer to §4.2 for formulae and empirical details.

and completeness require us to first compute importance matrix $M \in \mathbb{R}^{|\mathbf{f}|\times|\mathcal{Z}|}$. For example, if we train a multinomial probe to predict concept $z_j$ from feature $\mathbf{f}_i$, we can compute the importance of each dimension of $\mathbf{f}$ post hoc. Each concept $z_j$ defines a column of $M$. Note that $\forall i,j : M_{ij} \geq 0$, and $\sum_{i=1}^{|\mathbf{f}|} M_{ij} = 1$.

**Disentanglement** measures the average number of concepts $z_j$ that are captured by any single feature $\mathbf{f}_i$. To compute it, we first compute the entropy $H_\mathcal{Z}(P_{i\cdot})$ of the distribution $P_{i\cdot}$ defined over row $i$ of $M$: $P_{ij} = \frac{M_{ij}}{\sum_k M_{ik}}$. Disentanglement is then defined as $D_i = 1 - H_K(P_{i\cdot})$. This score is maximized when feature $\hat{\mathbf{f}}_i$ is only responsible for predicting a single concept $z_j$; it is minimized when feature $\mathbf{f}_i$ is equally important for predicting all concepts.

**Completeness** measures the average number of features $\mathbf{f}_i$ that are useful in predicting a single concept $z_j$. This score is defined analogously to disentanglement, but over columns $j$ in $M$: we take $C_j = 1 - H_\mathbf{f}(P_{\cdot j})$. Completeness is maximized when only one feature $\mathbf{f}_i$ is helpful in predicting $z_j$, and it is minimized when all features are equally important in predicting the concept.

**Informativeness** is inversely proportional to the prediction error of a probe trained on the feature vector. In the implementation of Eastwood et al. (2023), it is simply defined as the accuracy of a probe in predicting concept $z_j$ when trained on the feature vector $\mathbf{f}$. This captures whether a ground-truth concept is recoverable from the feature vector.

**Explicitness** is conceptually related to informativeness. $E$ captures the trade-off between the probe's capacity and the probe loss; this is measured as one minus the normalized area under the loss-capacity curve (AULCC); we refer readers to Eastwood et al. (2023) for details. This score is maximized when the lowest-capacity probe achieves the best loss, and thus that no excess capacity was required to fully recover a given concept.

### D.3 Further Details on Independence and Disjointness

To illustrate the conceptual distinction between independence and disjointness, we present diagrams in Figure 7. Intuitively, disjointness implies that two feature representations exist in non-overlapping subspaces of the model representations, and thus that the effect of steering both features can be predicted from the result of steering either in isolation. Conversely, independence implies that steering with one concept would not affect how the model uses other concepts. Refer to §4.2 for details.

## E Is One Dimension Sufficient?

In SAE-based interpretability studies, it is common to steer with a single feature, regardless of how many features receive high attributions for a given task. This corresponds to the following assumption:

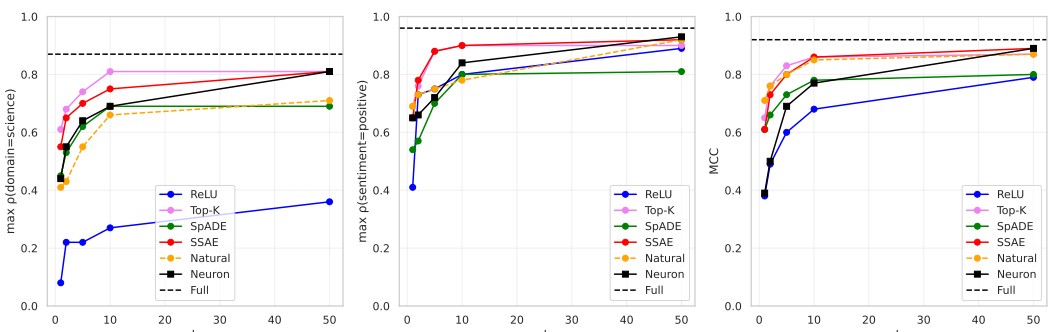

Figure 8: **Correlation coefficients between probe logits and concept labels for domain=science (left), sentiment=positive (middle), and MCC (right)**. Results for Gemma-2-2B shown here; results for Pythia-70M are in App. H. We vary the number of dimensions $k$ that the probe is allowed to have non-zero weights from. $k$-sparse probes trained on SAEs begin to converge around 10 dimensions for Top-K, SpADE, SSAE, and Natural, and recover most of the performance of a non-sparse probe that is allowed to use the entire residual vector (Full). $k$-sparse probes trained on the residual stream (Neuron) require more dimensions to converge, as expected.

**Assumption 2: One feature dimension is sufficient for concept detection and control.** *Given binary concept $z_i$ and feature vector $\mathbf{f}$, one dimension $\mathbf{f}_i$ of $\mathbf{f}$ is sufficient to represent and control $z_i$ in $\mathcal{M}$.*

To evaluate the extent to which this assumption holds in practice, we train $k$-sparse probes (as operationalized in Gurnee et al. (2023)) on featurized representations $\mathbf{f}$. $k$-sparse probes are linear probes that may have non-zero weights from up to $k$ dimensions of the representations they are trained on. Lachapelle et al. (2023a) established a connection between disentanglement and sparse prediction: they prove that disentanglement leads to optimal loss using sparse predictors. Further, as features become more entangled, we need to reduce sparsity regularization to maintain accuracy; this theoretical finding further motivates the following experiment.

**Hypothesis.** More dimensions yield monotonically increasing expressive power. Thus, performance should be non-decreasing as $k$ increases. We care primarily about when increasing $k$ begins to yield diminishing improvements in the MCC. Representations obtained with strong sparsity constraints, like SAEs, should reach this saturation point at smaller $k$ than representations with no such constraints, such as residual vectors.

**Results.** We display the (M)CC of $k$-sparse probes trained on feature vectors $\mathbf{f}$ in Figure 8. Top-K SAEs and SSAEs achieve the best trade-off between MCC and sparsity at all $k$; they also approach the MCC of training a normal probe on the full activation vector at the residual stream. ReLU SAEs do not begin saturating even at 10–50 features, whereas all other SAEs do. SSAEs and Top-K SAEs achieve better concept recovery at the same $k$ as the residual neuron baseline, whereas ReLU SAEs do not.

These results suggest that SAEs do confer sparsity benefits compared to the original activation space of $\mathcal{M}$, but also that one-dimensionality assumptions may often be insufficient—even when the concepts are relatively simple.

## F  PROBE ACCURACIES

Here, we present the accuracies of each probe we use in our disentanglement experiments and evaluations. We present these as heatmaps to verify whether each probe learn an independent

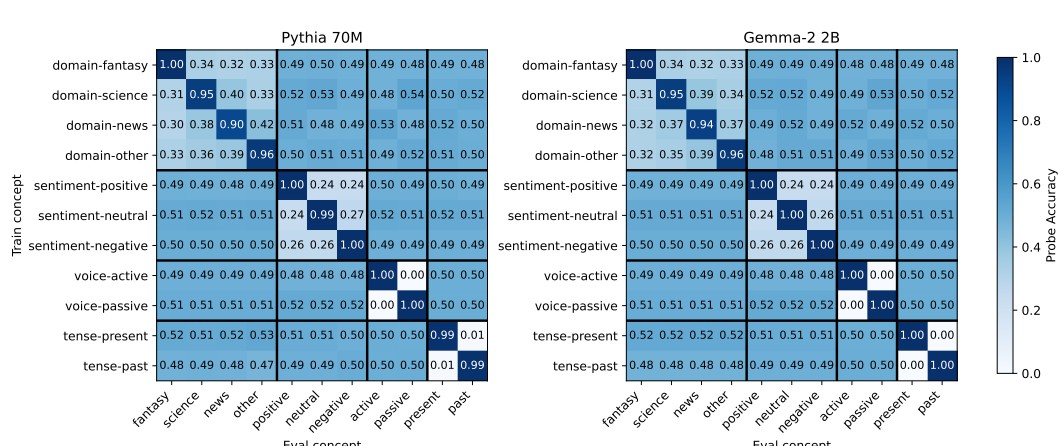

Figure 9: Accuracy of binary probes (rows) on all concept value classification tasks (columns). We expect high values on the diagonals, below random chance for within-concept value pairs, and random chance for across-concept value pairs.

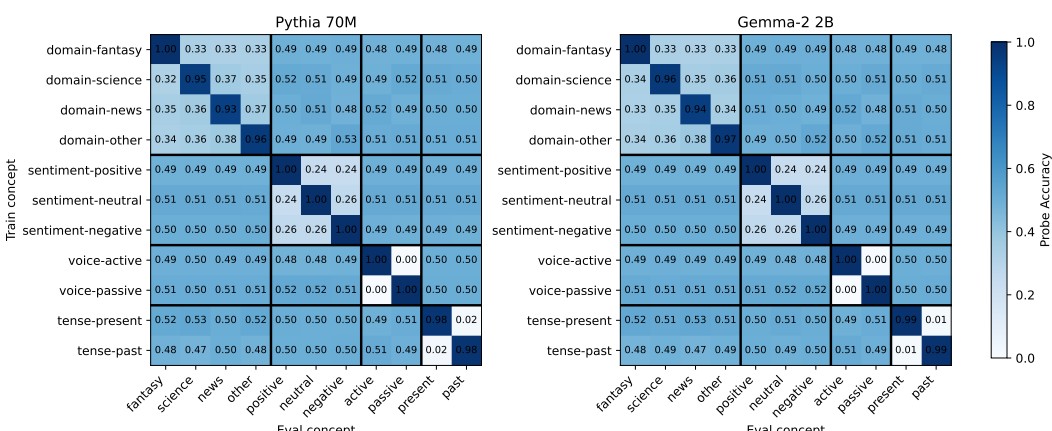

Figure 10: Accuracy of multinomial probes on all concept value classification tasks (columns). We expect high values on the diagonals, below random chance for within-concept value pairs, and random chance for across-concept value pairs.

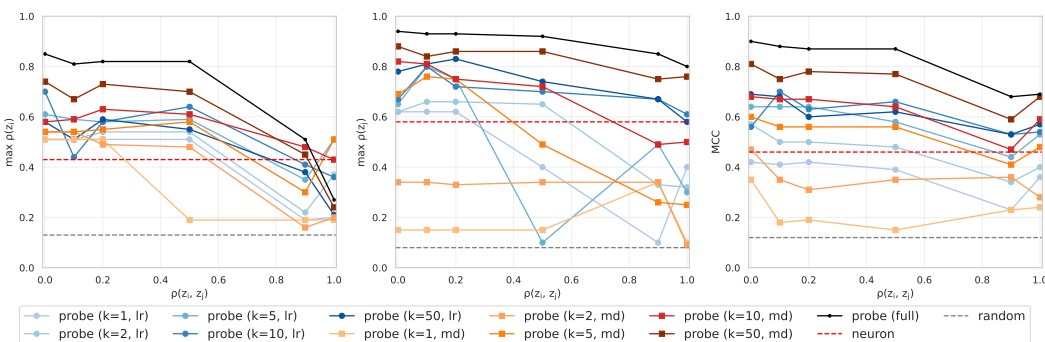

Figure 11: **MCC for the two most performant sparse probing methods from Gurnee et al. (2023) at various** $k$. Results for Pythia-70M shown here. The LR method achieves higher MCC at lower $k$, but MD overtakes LR at higher $k$.

representation of its target concept; if it does, we expect high scores along the diagonal, lower-than-random scores for within-concept pairs,[10] and random-chance scores for across-concept pairs.

Binary linear probes trained on the middle layers of Pythia-70M and Gemma-2-2B (Figure 9) achieve near-perfect accuracies on their respective concepts, and achieve the expected random accuracies on all other concepts. This empirically supports Assumption 1, and supports the idea that the MCC ceiling should be high (§3.1).

In §4.1 and §4.2, we instead use multinomial linear probes trained on the final layers of Pythia-70M and Gemma-2-2B. We find (Figure 10) that these probes also achieve the expected high accuracies on the target concepts, below-random-chance accuracies on within-concept pairs, and random-chance accuracies on across-concept pairs. This validates that the non-independence we observe in our steering experiments are not due to the probes, but rather are more likely due to the featurization methods that we use to steer.

## G  SPARSE PROBING

Here, we replicate the setup of Gurnee et al. (2023) in our cross-concept correlation setting. We aim to assess which $k$-sparse probing methods are more robust to cross-concept correlations at multiple $k$. We focus on the two most performant methods from Gurnee et al. (2023): max mean difference (MD), and logistic regression (LR). MD works by computing the average difference in activations between positive and negative samples, and taking the $k$ neurons whose mean activation difference is greatest. LR works by first training a logistic regression probe with $L_1$ regularization on the full activation vector, and then taking the top $k$ according to the weights of the probe.

We observe (Figure 11) that the logistic regression (LR) method of selecting neurons is more effective at lower $k$. Between $k = 5$ and $k = 10$, MD generally overtakes LR in performance. As we are more concerned with low-dimensional concept recovery, we focus on LR in the feature dimensionality experiment (§E).

## H  FURTHER DISENTANGLEMENT RESULTS

Here, we present correlation coefficients and MCCs for $k$-sparse probes trained with varying $k$ on SAEs for Pythia-70M. As with Gemma-2-2B, correlation coefficients tend to converge at around 10 dimensions; this suggests that the one-dimensionality assumption may not often hold in practice, even for much smaller models. Note also that the neuron baseline is far more performant for Pythia than Gemma; perhaps this is because $k = 10$ represents a far greater proportion of the dimensions of $\mathbf{h}^\ell$ for Pythia than Gemma. Other trends are largely consistent with Figure 8.

---

[10]We expect lower-than-random scores for within-concept pairs because a classifier trained on an alternative value of a concept should be strictly worse than a random probe, as the target label will be *negatively* correlated with the target concept.

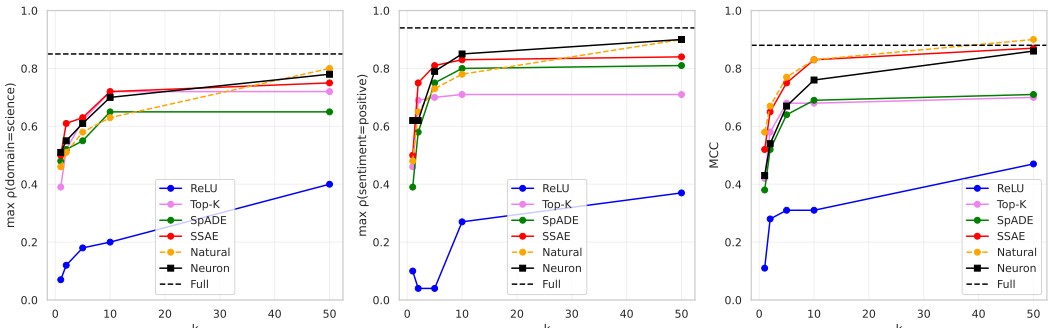

Figure 12: **Correlation coefficients between probe logits and concept labels for domain=science (left), sentiment=positive (middle), and MCC (right)**. Results for Pythia-70M. We vary the number of dimensions $k$ that the probe is allowed to have non-zero weights from. As with Gemma-2-2B, correlation coefficients tend to converge at around 10 dimensions. However, the neuron baseline is far more performant; perhaps this is because $k = 10$ represents a far greater proportion of the dimensions of $\mathbf{h}^\ell$ for Pythia than Gemma. Other trends are largely consistent with Figure 8.

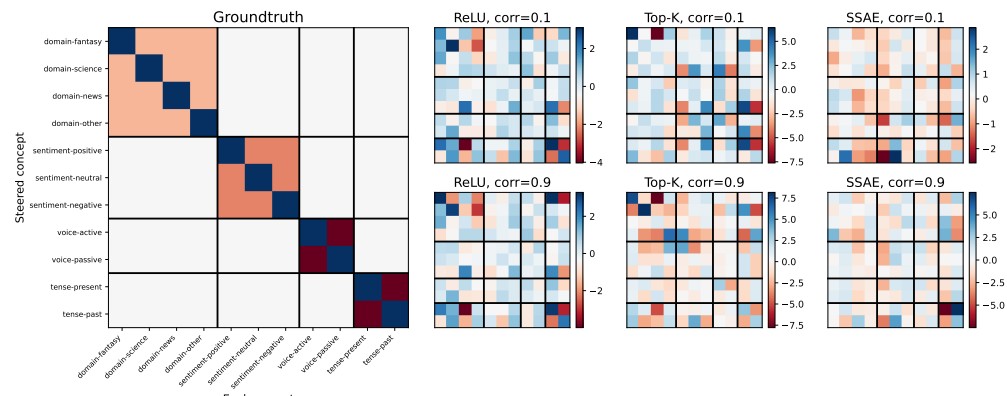

Figure 13: **The effect of steering a given concept (row) on the logit of another (column), as measured by a probe**. Results for Gemma-2-2B. If concept representations are causally independent, we expect a heatmap that resembles the ground-truth: $\Delta$LOGODDS should be high on the diagonal, negative for within-concept pairs, and close to 0.0 for across-concept pairs. All SAEs demonstrate the expected diagonals, but also significant across-concept effects, indicating non-independence. Increasing correlations in the training data, even up to 0.9, do not significantly change the trends.

# I FURTHER STEERING RESULTS

Here, we present steering heatmaps for Gemma-2-2B (Figure 13). Features appear less independent than for Pythia-70M, as indicated by more significant across-concept $\Delta$LogOdds for many concept pairs. That said, the expected diagonal trend is still present. This is further evidence that SAE features do not often correspond to causally independent concept representations.

We also present more detailed multi-feature steering results (Figure 14). We observe that features are often entirely disjoint (the two purple lines almost always completely overlap) while not being independent (the red line is not always perfectly flat at 0.0). We observe some distinction between the predicted and actual $\Delta$LOGODDS for ReLU SAEs, indicating that their affected subspaces do overlap slightly; this provides some evidence that disjointness is not tautologically expected—that is, a well-trained SAE can achieve it, but it is not guaranteed. This underscores that even when SAEs learn disjoint representations, one cannot use this as a proxy for causal independence.

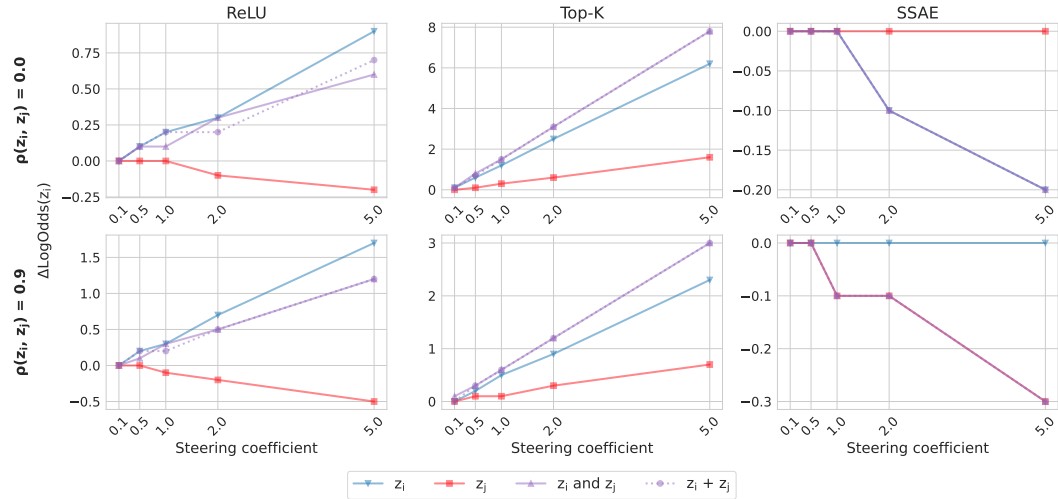

Figure 14: $\Delta\text{LOGODDS}(z_i)$ **under various steering coefficients** $\alpha$ **for steering feature** $\hat{\mathbf{f}}_i$**, and** $\hat{\mathbf{f}}_j$ **for a different concept** $z_j$**.** Here, concept $z_i$ is domain=science, and $z_j$ is sentiment=positive. Results for Gemma-2-2B. $\hat{\mathbf{f}}_i$ and $\hat{\mathbf{f}}_j$ are more often independent here than for Pythia-70M, as indicated by flat red lines. $\hat{\mathbf{f}}_i$ and $\hat{\mathbf{f}}_j$ are typically nearly disjoint, as indicated by the dotted purple line and solid purple lines almost (but not completely) overlapping.

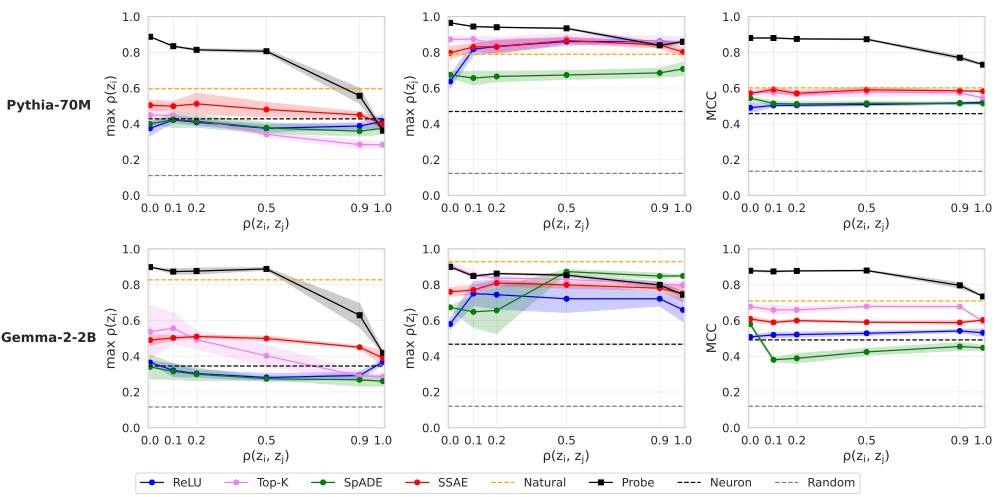

Figure 15: **Maximum correlation coefficient for tense=past (left), voice=passive (middle), and MCC (right) under varying correlational conditions.** Shaded regions represent 1 std. dev. across 3 training seeds. Ideal performance looks like a flat line at a high MCC. Probes perform best, and Top-K SAEs are best among unsupervised methods.

## J  ADDITIONAL VARIABLE CORRELATION EXPERIMENTS

The experiments thus far have focused on correlations specifically between the science domain and positive sentiment. To assess how well these results generalize to new variable correlations, we rerun our experiments while instead correlating the past tense with the passive voice.

We first present MCC results (Figure 15). Findings are largely consistent with Figure 2: supervised featurizers like probes perform best by far, but their performance drops sharply from $\rho$=0.9. Top-K SAEs are still the best-performing methods among unsupervised featurizers given our data. One difference here is that SAEs trained on large-scale natural language corpora are far better at recovering tense=past than our SAEs—especially for Gemma-2-2B.

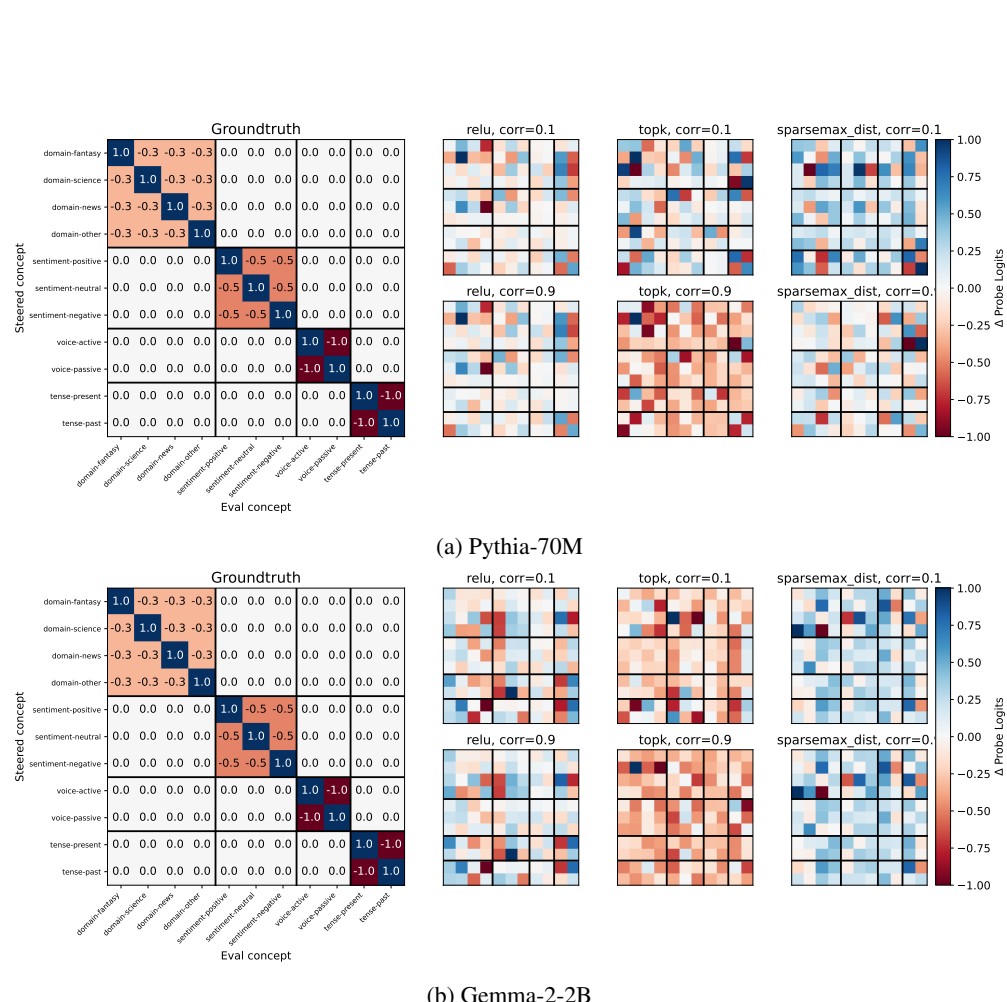

(a) Pythia-70M

(b) Gemma-2-2B

Figure 16: **The effect of steering a given concept (row) and the logit of another (column), as measured by a probe.** Results for Pythia-70M (a) and Gemma-2-2B (b). As when training on the previous set of correlated variables, diagonals are largely present, as are within-concept effects. There are significant across-concept effects, however, indicating non-independence.

Results when steering (Figure 16) largely resemble the previous results, especially for ReLU and SpADE SAEs. However, Top-K SAEs are now far more likely to demonstrate negative across-concept effects. This negative relationship might be expected for mutually exclusive concepts, but not if concepts are truly independent of each other. This supports our prior claim that existing unsupervised methods may demonstrate decent recovery of known concepts (as indicated by correlational metrics), but simultaneously produce entangled effects on multiple concepts (as indicated by interventional metrics).

## K  QUALITATIVE EXAMPLES OF STEERING

Here, we show examples of model generations before and after steering the top feature for "domain=science". We select the top feature using the same method as in §4.1: we use gradient attributions to a binary probe trained to predict a positive label if the input sentence is about science.

| Pythia-70M, ReLU | | |
| --- | --- | --- |
| It has been found that | | |

| No steering | Sentiment=positive ($\rho = 0$) | Sentiment=positive ($\rho = 1$) |
| --- | --- | --- |
| the first person to be the one who is the one who is the one | the most important part of the process of the process of the process of the | the presence of a high-fidelity material in the air-conditioning system is a very important factor |

| Gemma-2-2B, Top-K | | |
| --- | --- | --- |
| Once upon a time, | | |

| No steering | Domain=science ($\rho = 0$) | Domain=science ($\rho = 1$) |
| --- | --- | --- |
| a time when my children were very small, I bought a box of pencils (yes, that time). The box bore a very clear message: "It's never | there was a little girl who was born with a rare genetic disorder. She was born with a condition called "congenital heart disease." This condition is a birth defect that affects the heart's structure and function. | there was a brave little girl who was born with a heart condition. She was born with a hole in her heart, |

## L  LLM USAGE

The authors used large language models primarily as a polishing tool during writing. LLMs were not used in a significant capacity for writing experimental code nor for research ideation, although we acknowledge that libraries on which our code was based may have used LLMs.

