# OpenReview forum: "From Isolation to Entanglement: When Do Interpretability Methods Identify and Disentangle Known Concepts?"
_ICLR.cc/2026/Conference — ICLR 2026 Conference Withdrawn Submission_

### Official Review · Reviewer_7K1D · 2025-10-27

**Soundness:** 3
**Presentation:** 3
**Contribution:** 3
**Rating:** 4
**Confidence:** 3

**Summary:**

This paper evaluates Sparse Autoencoders (SAEs) on their ability to find features which "disentangle" concepts, as measured by 3 criteria:

1) independent manipulability, meaning steering with a disentangled feature affects only the one associated concept, where the authors find all SAEs to be poorly disentangled;
2) sparse prediction, meaning only one (or a few) features are needed to predict the presence of the concept, where the authors find all SAEs predict concepts better with k>1 features than the maximally-sparse k=1 predictor, and ReLU-based SAEs require relatively more features;
3) disjointedness, meaning that multiple steerings with different features should combine their effects additively, which the authors find across all SAEs and language models.

These results come from three architectures of SAEs trained by the authors on two publicly-available language models.

**Strengths:**

The paper addresses an important lacuna in the field of interpretability research. Prior research has focused on evaluating features in isolation, but does not consider their interactions with other features.

The paper has excellent presentation, clearly defining how it seeks to measure disentanglement, setting lower and upper bounds with the "baselines and skylines" section, and helping the reader understand the shape of its figures in the text and the "groundtruth" in Figure 4.

**Weaknesses:**

The paper is modest in scope. It seeks to evaluate existing techniques via new metrics, but without making the evaluations into a benchmark which could be easily used and built upon by other researchers.

Despite otherwise clear communication, the paper omits otherwise key information to reproduce its results. This includes:
- Which layer(s) of the language model were used to train the SAEs.
- Hyperparameters of SAE training, such as number of features and amount of sparsity.
- The structure of the sentence data described in lines 126-132 (see question below).
- Which tokens in the sentences were used for training and evaluating the SAE features.

The paper's Section 4.2 is potentially exciting, but the results depicted in Figure 5 are not a sufficient test of the proposed hypothesis. In particular:

- Figure 5 appears to be results from a particular choice of concepts/features z_i and z_j. The reader is not informed which features were used to make Figure 5, and is given no evidence that this behavior generalizes to other features.

- These results are only shown via visual inspection of Figure 5, but the different magnitudes of effects make this hard to assess. For instance, in the top-middle figure, intervening on z_i produces apparently-negligible shifts in the logit for z_i, while intervening on z_j or {z_i and z_j} dramatically suppresses the prediction of z_i. But because the effect of intervening on z_j appears >100x larger than the effect of intervening on z_i, it is difficult to assess whether there is truly additivity for the combined intervention. For instance, if the z_j intervention has a -1000 logit effect, and the z_i intervention has a +1 logit effect, Figure 5 is not able to show whether the combined effect is a -999 logit effect (additive) or a -1010 logit effect (not additive).

Both of these critiques could be addressed by moving Figure 5 to an appendix and creating a new figure in its place. This new figure would be like Figure 4, showing a single numeric measurement for the interaction of each pair of concepts. The numeric measurement would aim to capture the nonlinearity of the interaction, such as LogOdds(z_i| steered by z_i,z_j)+LogOdds(z_i|no steering)-LogOdds(z_i|steered by z_i)-LogOdds(z_i|steered by z_j), with suitable normalization.

**Questions:**

Can you provide examples of the sentence data, as produced in lines 126-132? Are they just the abstract labels (e.g., "active, past, negative, fantasy"), full sentences (e.g., "the dark lord Sauron deceived them"), or something else? A short appendix of example sentences would make it easier to understand what data was used for these tests.

In Figure 5, some graphs have only the purple+red lines or the purple+blue lines. Is this because the purple line completely covers a red/blue lines? If so, please include a comment to that effect in the caption.

On line 242, the paper says "The gap between SAE architectures is significant and consistent across models. Top-K in particular performs well." This is not reflected in Figure 2. TopK SAEs seem to excel in the left images (domain=science), but are on par or worse than other architectures in the other four images. Is this a mistake, or based on some other analysis?

---

> ### Author Response · Authors · 2025-11-21
> **Response to 7K1D**
>
> We appreciate the detailed suggestions for improvement. We have made the following changes to our submission (more detail below):
> * Additional training details and data generation details in Appendices A and B
> * Clarified Fig. 5 and included additional multi-concept--steering experiments with the sentiment=positive concept in Fig. 14 in the Appendix
> * Clarified our statement about Fig. 2
>
> > The paper is modest in scope. It seeks to evaluate existing techniques via new metrics, but without making the evaluations into a benchmark which could be easily used and built upon by other researchers.
> Our primary goal is to bridge mechanistic interpretability and causal representation learning (CRL) by bringing evaluation techniques from both into one paper. Specifically, how well do identifiability metrics from CRL predict success in common interpretability applications, like steering—and vice versa? Surprisingly, the two do not seem strongly predictive of each other.
>
> The data and CFG generation code will be released, as will our evaluation code. We believe that future work could build on this by adding additional concepts. On that note, we acknowledge that the addition of a greater diversity of concepts could always improve the robustness of evaluations of this sort. Our data provides partial ecological validity while allowing us to easily verify outputs, and also ensures that we do not dilute the correlational metrics when correlating one pair of concepts.
>
> > [T]he paper omits otherwise key information to reproduce its results. This includes which layer(s) of the language model were used to train the SAEs[, and] hyperparameters of SAE training, such as number of features and amount of sparsity.
>
> See **Appendix B** in the revised PDF; this section covers all of our training details, including hyperparameters, as well as test MSE and sparsity metrics.
>
> > The structure of the sentence data described in lines 126-132 (see question below).
>
> See **Appendix A** in the revised PDF. These are natural language sentences generated by a PCFG with attributes corresponding to our generative factors.
>
> > Which tokens in the sentences were used for training and evaluating the SAE features.
>
> Our training setup and implementation follows that of [1]. **We uniformly sample token representations from a large dataset, excluding special tokens like [PAD] tokens, batch these representations, and use these batches to train our SAEs.**  For computing MCCs, we compute the token-wise correlation between feature activations and the presence of the ground-truth concepts—that is, we include all tokens in the dataset (except special tokens). Thus, there is a close match between the settings in which the train and test token representations are obtained.
>
> > Figure 5 appears to result from a particular choice of concepts/features z_i and z_j. The reader is not informed which features were used to make Figure 5, and is given no evidence that this behavior generalizes to other features.
>
> The results were for one of the concepts correlated in the experiments in Sec. 3 (domain=science); we acknowledge that this was unclear. As part of a larger effort to clarify the presentation of results, we have reformatted Figure 5. This included using the name of the concepts we were measuring, and plotting predicted vs. actual $\Delta$(LogOdds) instead of $\Delta$(LogOdds). Note that this figure now includes results for two concepts (domain=science and sentiment=positive) instead of one.
>
> We have since run additional disentanglement and steering experiments **using the other two concepts, voice and tense**, and included them in Appendix J in Figs. 15 and 16. Trends are largely the same as before.
>
> > These results are only shown via visual inspection of Figure 5, but the different magnitudes of effects make this hard to assess.
>
> We agree that a quantitative metric would be helpful. To quantify independence and disjointness, we have proposed quantitative metrics that summarize Figures 4 and 5. See Tables 1 and 2 in Sections 4.1 and 4.2. These metrics are defined in Eq. 2, and are similar in spirit to the interventional robustness score and to normalized pointwise mutual information.
>
> > Can you provide examples of the sentence data?
>
> We have **included example sentences in the newly added Appendix A**; see Table 3. Here are some example sentences:
>
> ```
> Unsuccessfully, the malevolent dragon damaged the corrupted land. | voice=active, tense=past, domain=fantasy, sentiment=negative
>
> The brilliant scientist celebrates the remarkable findings. | voice=active, tense=present, domain=science, sentiment=positive
>
> The event was explained in the recent report. | voice=passive, tense=past, domain=news, sentiment=neutral
>
> The pleasant surprise is endorsed advantageously by the talented artist. | voice=passive, tense=present, domain=other, sentiment=positive
> ```

---

> > ### Author Response · Authors · 2025-11-21
> > **Response to 7K1D (pt. 2)**
> >
> > > In Figure 5, some graphs have only the purple+red lines or the purple+blue lines. Is this because the purple line completely covers a red/blue lines? If so, please include a comment to that effect in the caption.
> >
> > Yes, **this is because the lines overlap completely** (indicating perfect disjointness). The presentation of these results was a common point of feedback, so we have changed the format of this figure. Now, we instead plot predicted accuracy under perfect disjointness versus actual accuracy. If two features are disjoint, the $\Delta$(LogOdds) when steering two features should be exactly equal to the predicted $\Delta$(LogOdds). Indeed, we observe that this is the case most of the time in the new figure. We have also added a table where we compute the R^2 between the predicted and actual $\Delta$(LogOdds) as a more concise quantitative summary.
> >
> > > On line 242, the paper says "The gap between SAE architectures is significant and consistent across models. Top-K in particular performs well." This is not reflected in Figure 2.
> >
> > We said this because **the correlation coefficient for domain=science and the MCC for Top-K SAEs were significantly higher than for other SAE architectures**. While Top-K SAEs scored slightly worse than other SAEs for sentiment=positive, its average performance across all concepts was high.
> > We have since added another architecture, SSAEs, which are trained on differences between pairs of activations, instead of activations from individual examples. SSAEs perform even more strongly w.r.t. the MCC metric, as predicted by their theory [2]. Thus, the text now focuses more on SSAEs. Top-K SAEs still achieve the best MCC among SAEs trained on activations, rather than activation differences; we have revised the text to point to the specific evidence for our claims.
> >
> > # References
> >
> > [1] Hindupur et al. (2025). “Projecting Assumptions: The Duality Between Sparse Autoencoders and Concept Geometry.” https://arxiv.org/abs/2503.01822
> >
> > [2] Joshi et al. (2025). “Identifiable Steering via Sparse Autoencoding of Multi-concept Shifts.” ICML Actionable Interpretability Workshop. https://arxiv.org/abs/2502.12179

---

### Official Review · Reviewer_KYKH · 2025-10-30

**Soundness:** 1
**Presentation:** 1
**Contribution:** 2
**Rating:** 2
**Confidence:** 4

**Summary:**

- This paper studies the ability of sparse autoencoders (SAEs) and supervised probes to identify and disentangle concept representations when concepts are correlated or entangled.
- The authors create synthetic datasets based on natural language where they can manually control the correlation between concepts.
- They train SAEs and probes on model representations from datasets with varying levels of feature entanglement
- they then evaluate how well learned features capture the underlying concepts via correlation, sparse probing, and independence/disjointness metrics when steering.
- The results show that interpretability methods generally degrade as concept entanglement increases.

**Strengths:**

- Evaluating the ability of interpretability methods to disentangle correlated concepts is a very relevant question for the field. The approach of using synthetic datasets with controlled correlations is elegant and allows interpretability methods to be evaluated on actual models with still natural-ish text.
- Diverse evaluation learned representations - feature alignment, sparse probing, steeringto measure via independence and disjointness
- Diverse set of interpretability methods and baselines evaluated, including "organic" SAEs (trained on normal text), individual neurons, and supervised probes.

**Weaknesses:**

- Several load-bearing details are omitted from the manuscript. It's unclear how sampled concepts are used to construct the actual natural language sentences in the dataset. For SAEs, critical training details are missing—particularly sparsity levels and reconstruction loss. Without these metrics, it's impossible to determine whether SAE performance differences reflect poor training or actual effects of data correlations.
- The synthetic data setup is compelling, but the experiments are limited to just four categorical features, which may not capture the complexity of real concept entanglement.
- Many experimental results feel poorly presented and would benefit from more curation:
  - The sparse probing results (Figure 4) only show the effect of increasing k (number of features in the probe). It's unsurprising that performance increases with k. To align with the paper's main question, these results should show performance as a function of data correlation in the training dataset.
   - The steering results rely heavily on qualitative heatmap inspection. Aggregate quantitative metrics would be more convincing than requiring visual inspection.
  - The presentation around disjointness is unclear. The delta log odds magnitudes in Figure 5 seem problematically large (values around 4,000)—this would imply steering increases concept probability by e^4000, suggesting possible numerical issues or extremely low base rates.

**Questions:**

- Could you provide the missing SAE training details—specifically sparsity levels and reconstruction losses across different training conditions? This would help determine whether performance differences are due to training quality or data correlation effects.
- How are natural language sentences constructed from the sampled synthetic features? Could you provide qualitative examples of sentences in your dataset?
- Can you clarify the disjointness results in Figure 5? The delta log odds values (~4000) seem implausibly large—are there numerical stability issues, or is this due to extremely low base rates?

---

> ### Author Response · Authors · 2025-11-21
> **Response to KYKH**
>
> We appreciate the detailed suggestions for improvement. We have made the following changes to our submission (more detail below):
> * Additional training details and data generation details in Appendices A and B
> * Quantitative metrics for the steering results (selectivity, steering independence, and R^2)
> * Clarified the purpose of the sparse probe experiments
> * Identified the reason behind the large range of Delta(LogOdds) as the large norm change in the Top-K SAE features and added qualitative analyses with a reduced steering coefficient (App. J)
>
> > It's unclear how sampled concepts are used to construct the actual natural language sentences in the dataset.
>
> We have added further details in the revised PDF; see **Appendix A** for details about how we use these concepts to construct a CFG that generates natural language sentences. We also provide examples of generated sentences.
>
> > For SAEs, critical training details are missing—particularly sparsity levels and reconstruction loss. Without these metrics, it's impossible to determine whether SAE performance differences reflect poor training or actual effects of data correlations.
>
> We have added a section to the appendix clarifying these details; see **Appendix B** in the revised PDF.
>
> > The experiments are limited to just four categorical features, which may not capture the complexity of real concept entanglement.
>
> We agree, and now acknowledge this limitation in the main text. We focused on four distinct but relatively simple concept types (syntactic/semantic, binary/multinomial) to ensure the ground-truth labels remained rigorous and verifiable. While broader diversity is a goal for future work, this controlled scope allows for precise quantification of disentanglement and the selectivity of steering operations. We believe that the findings across these concepts are consistent enough to support our main claims.
>
> > It's unsurprising that performance increases with k. To align with the paper's main question, these results should show performance as a function of data correlation in the training dataset.
>
> It is indeed unsurprising that performance increases as a function of k; the surprising part is that **the MCC difference is so large between k=1 and larger k values**. If an SAE truly localizes concepts in a sparse manner, it should not take so many dimensions to represent binary concepts. Prior notions of disentanglement conceptualize it as a one-to-one mapping from representation dimensions to concepts [1,2]. This suggests that a common basic motivation for deploying SAEs—that they can localize concepts concisely—may not actually be met. (That said, they are still significantly better than the original activation space.)
>
> Given the findings in Figure 2, we believe that these results would look largely the same up to correlations of about 0.9, at which point all MCCs would decrease at all k.
>
> > The presentation around disjointness is unclear. The delta log odds magnitudes in Figure 5 seem problematically large (values around 4,000)—this would imply steering increases concept probability by e^4000, suggesting possible numerical issues or extremely low base rates.
>
> We have investigated this anomaly and identified a critical structural difference between architectures. Top-K SAEs lack the norm constraints of L1-regularized SAEs, and this caused the norms of the steered activations to explode during steering. Because the probe is linear, its logits scale proportionally to increases in activations in the direction defined by the probe’s weights. For all other SAEs, the steered norms generally increase, but their norm increases remain in the low hundreds—a much more reasonable activation range.
>
> We have now reduced the steering coefficient to 0.5 for Top-K and rerun all of our Top-K SAE steering experiments, including in the new tables; this has produced Delta(LogOdds) in much more reasonable ranges.  As a qualitative validation of our current steering settings, when steering the model during free-form generation, the previous Top-K steering coefficient caused the model to immediately stop generating. After reducing the steering coefficient to 0.5 for Top-K, we obtain more reasonable model generations that seem to correspond to the target concepts (see Appendix K). We have also validated the previous steering coefficient for ReLUs and obtained similar outputs. We plan to add additional examples to this appendix.

---

> > ### Author Response · Authors · 2025-11-21
> > **Response to KYKH (pt. 2)**
> >
> > > The steering results rely heavily on qualitative heatmap inspection. Aggregate quantitative metrics would be more convincing than requiring visual inspection.
> >
> > We agree that a better quantitative metric would help summarize these results more concisely and convincingly. To summarize the steering results, we now propose two metrics inspired by the worst-case IRS scores for evaluating steering: feature selectivity and steering independence. Selectivity measures whether a feature steers only the target concept (and is computed over a row of the steering heatmap). Steering robustness measures whether a given concept is only affected by its respective feature (and is computed over a column of the steering heatmap). See the updated Section 4, as well as Table 1. Our results indicate that metrics which capture input-to-feature relationships do not sufficiently explain feature-to-output relationships: there is widespread non-independence and non-selectivity.
> >
> > We believe all questions have been addressed via addressing the weaknesses. We would be happy to continue the discussion if any questions remain!

---

### Official Review · Reviewer_HZt6 · 2025-11-02

**Soundness:** 3
**Presentation:** 3
**Contribution:** 2
**Rating:** 4
**Confidence:** 3

**Summary:**

This paper studies the disentanglement of features-extractors like SAE on a variety of language models.
The authors show, that depending on the method and number of $k$ components used to isolate a single ground-truth concept, SAE-like models tend to vary their disentanglement as measured with MCC, while still relatively underperforming probes competitors.
Overall, this paper highlights the importance of data through in-distribution correlations between concepts when performing dictionary learning and the number of components in use to extract disentangled information from latent representations.

**Strengths:**

The idea of measuring disentanglement and so interpretability of SAE-like methods is timely and important to gain insight on what we can expect in practice for these methods. The insight about steering and disjointness are useful and well-explained in the paper.

The paper is well presented, and research analyses are well formulated and investigated with rigor.

Overall, the paper shows in a clear way that SAE-like methods do not come with guarantees for properly disentangling ground-truth concepts on their experimental data, and this analysis complements previous findings that many SAE-like methods do not come with guarantees about recovering interpretable features.

**Weaknesses:**

This paper misses comparisons to other works that recently appeared in SAE literature and treat similarly related aspects (among which identifiability), e.g. [1,2,3].  For this, I cannot say the paper excels in novelty.

Also, while MCC is a quite popular metric for studying disentanglement of representations, it has some pitfalls (since it only tests correlations) that other disentanglement metrics cover, see e.g. [4,5]. For example, DCI-ES [5] includes a training phase on a probe to detect which SAE components affect the prediction. The final disentanglement score accounts for the sparsity of the probe on SAE features.
IRS [6] instead considers do operations within a causal framework identical to the one considered in this paper. \
Having these two metrics would be desirable for a much more in-depth analysis of disentanglement, and if not integrated, the authors should at least mention them and explain which similar aspects would be detected by using them (for example, if a feature depends on two concepts it would give a quite low IRS, because of lack of interventional independence).

Furthermore, I found the details about the data generation vague, the presentation should further explain how data are generated and used to train the SAEs beyond Natural baselines. In this respect, the disentanglement analysis is limited to the synthetic dataset the authors use, and conclusion about disentanglement of SAEs in all, in-distribution test data for language models cannot be easily provided.

-----

[1] Are Sparse Autoencoders Useful? A Case Study in Sparse Probing, Kantamneni et al. 2025 \
[2] Identifiable Steering via Sparse Autoencoding of Multi-Concept Shifts, Joshi et al. 2025 \
[3]  On the Theoretical Understanding of Identifiable Sparse Autoencoders and Beyond, Cui et al. 2025 \
[4] Measuring Disentanglement: A Review of Metrics, Carbonneau et al. 2020 \
[5] DCI-ES: An Extended Disentanglement Framework with Connections to Identifiability, Eastwood et al, 2023 \
[6] Robustly Disentangled Causal Mechanisms: Validating Deep Representations for Interventional Robustness, Suter et al., 2019

**Questions:**

I don't have specific questions for the authors, but I hope to see some discussion on the weaknesses.

---

> ### Author Response · Authors · 2025-11-21
> **Response to HZt6**
>
> We appreciate the detailed suggestions for improvement. We have made the following changes to our submission (more detailed responses below):
> * Additional SSAE experiments and citations in recent work on SAEs
> * Clarified that we need different techniques to evaluate features that locate concepts, and those that enable steering
> * Adopted DCI-ES scores and highlighted the shortcomings of the MCC in the related work
> * Acknowledged the limitations of our synthetic experiments
> * Additional training details and example data points in Appendices A and B
>
>
> > This paper misses comparisons to other works that recently appeared in SAE literature and treat similarly related aspects (among which identifiability)
>
> Thanks for pointing us to these resources. We have integrated these into the paper where relevant. Note in particular the inclusion of the DCI-ES metrics in Sec. 3, as well as novel steering metrics inspired by the interventional robustness score in Sec. 4.
>
> We have also **added Joshi et al.’s SSAE method** [1] to our analyses. As predicted by their theory, this method outperforms typical SAEs on MCC (but interestingly, not on DCI metrics). Maximal steering independence and selectivity scores also look better than for other architectures, although the means are similar.
>
> > While MCC is a quite popular metric for studying disentanglement of representations, it has some pitfalls (since it only tests correlations) that other disentanglement metrics cover
>
> We agree. Part of our motivation was to highlight the pitfalls of using correlational and observational metrics like MCC. But as our steering experiments show, high observational metrics are not sufficient for establishing robust disentanglement under interventions to latents.
> Inspired by the IRS metric, **we have proposed metrics to summarize the results of the steering experiments more concisely**; see Tables 1 and 2 in the updated manuscript.
>
> **Our primary motivation is to highlight how techniques from mechanistic interpretability can help improve causal representation learning, and vice versa.** CRL papers tend not to consider steering (although [1], whose method we now include, is a notable exception), whereas mechanistic interpretability studies tend not to directly evaluate disentanglement. We have aimed to make this motivation clearer in the revision; thank you for raising this point.
>
> >  I found the details about the data generation vague, the presentation should further explain how data are generated and used to train the SAEs beyond Natural baselines.
>
> To address this, we have added further details to the revised manuscript:
> * See **Appendix A** for information about the data, including the CFG and some generated examples. In particular, see Figure 6 for excerpts from our CFG, and Table 3 for examples of generated sentences.
> * See **Appendix B** for information about the training procedure, including hyperparameters, number of examples, among other details.
>
> > The disentanglement analysis is limited to the synthetic dataset the authors use, and conclusion about disentanglement of SAEs in all, in-distribution test data for language models cannot be easily provided.
>
> This is indeed a limitation. We have added an acknowledgement of this in the updated manuscript.
>
> # References
>
> [1] Joshi et al. (2025). “Identifiable Steering via Sparse Autoencoding of Multi-concept Shifts.” ICML Actionable Interpretability Workshop. https://arxiv.org/abs/2502.12179

---

> > ### Comment · Reviewer_HZt6 · 2025-11-24
> > **Reply to authors**
> >
> > Thank you for including a new method and DCI-ES.
> >
> > **About steering.** The role of steering to me seems quite related to the fact that you can identify a factor, whose linear variations reflect into changes in behavior. In that sense, I don't see why it should differ from a notion of disentanglement. Can you elaborate on this?
> >
> > I would be curious to see the metric inspired by IRS.
> > If authors believe I should check new details, can you please highlight changes in a different colour (if OR permits updating the text of the submission)?

---

> ### Author Response · Authors · 2025-11-24
>
> Thanks for continuing the discussion!
>
> **On steering.** Imagine that we find a direction that, when steered, causes multiple concepts to change simultaneously. For example, steering with the domain=science direction could cause the model to generate noticeably more scientific sounding outputs, but it could also cause the model to generate more passive-voice sentences. This would be a case where we have successfully found a steering direction for domain=science, but also where this direction is entangled with another concept. Conversely, one could establish that two concepts have their own directions (i.e., are disentangled) in the representation space, but also that steering with either direction doesn't have any interpretable impact on model behavior (see Arad et al. (2025) [2] for a more detailed explanation of this phenomenon). Thus, steering and disentanglement are orthogonal: one can have any combination of either.
>
> Yes, OR does allow revisions! We have uploaded a revised PDF with the new metrics and appendices to this OR submission. Please see **Eq. 2 on page 7** for the new metrics inspired by IRS. Also see **Table 1 at the top of page 8** for results using this new metric. Note that this is distinct from IRS in that it operates on the output space after interventions to representations, whereas IRS operates on the representation space after interventions to the input concepts. (This entire section and others were refactored, so highlighted changes may not be particularly interpretable; that said, we would be happy to provide further pointers and guidance toward specific details to facilitate cross-checking!)
>
> [2] Arad et al. (2025). “SAEs Are Good For Steering - If You Select the Right Features.” EMNLP. https://aclanthology.org/2025.emnlp-main.519/

---

### Official Review · Reviewer_tsv2 · 2025-11-04

**Soundness:** 3
**Presentation:** 3
**Contribution:** 2
**Rating:** 4
**Confidence:** 4

**Summary:**

The paper's goal is to design a new evaluation of the representation disentanglement properties of modern sparse decomposition interpretability techniques such as sparse autoencoder (SAE) variants.

The authors use a synthetic dataset with ground-truth concept labels used to generate ~400k sentences of text (with ~1k sentences held out in a test set). The concepts, together with their discrete sets of possible values, are:
- voice (active, passive)
- tense (present, past)
- sentiment (positive, neutral, negative)
- domain (news, science, fantasy, other)
The authors can also control the degree of correlation between a given pair of concepts.

The paper proposes and carries out the following evaluations:
- compute the "mean correlation coefficient" (MCC) under increasing concept correlation between:
	- the presence of a binarized version of a ground-truth concept (so for concepts taking $n>2$ values, we create $n$ binarized concepts corresponding to each value) in the activation (we know this because we control the data generation process)
	- the activations of the SAE latent most correlated with the binarized concept
	- this is averaged over concepts, hence the "mean" in the name. Note that MCC would be more appropriately called "mean *max* correlation coefficient", because we first pick the SAE latent with max correlation.
	- results show that SAEs generally fare worse here compared to linear probes trained with supervision (where we use the probe's logits to compute MCC)
- compute the "mean correlation coefficient" (MCC) between:
	- the presence of the same binarized concepts from above;
	- the logits of a $k$-sparse probe trained on an SAE's latents for small-ish values of $k$ (to allow for concepts encoded in multiple SAE latents)
	- results show SAEs again fall short of probes, but the gap is smaller, and top-K SAEs get to within a few percent MCC with $k\approx10$.
- edit activations with the goal of changing the concept value encoded *without* changing the values of other concepts. Given an SAE, this works by finding the SAE latent with the highest gradient attribution to the logit of a binary probe trained on the last layer of the model (even though in this experiment the SAE is trained on activations from the middle layer). This latent is then used to do a steering-like manipulation on the activation. The logits of a probe trained to predict the concept in the activation space the SAE operates on are used to evaluate the success of this intervention.
	- Results show that activation edits work to change the targeted concepts, but have many side effects
- test for "disjointness" of concept representations, which is defined as checking if steering with two SAE latents simultaneously will result in the same change of log-odds for a concept (as judged by a probe) as steering one and then the other.
	- Results show almost perfect disjointness. However see weaknesses.

**Strengths:**

Overall, the paper shows that SAEs struggle compared to supervised methods along several interesting axes, and may need multiple latents to express a single concept. Other strengths include:
- a controlled setup that is nontrivial enough to be interesting while still retaining useful ground truth
- lots of relevant experiments
- writing is clear for the most part, though there are a lot of details to the experiments.

**Weaknesses:**

- There is only a synthetic dataset evaluation, and the work feels overall incremental compared to prior works.
- there are lots of different procedures to assign SAE latent(s) to a concept (at least three by my count), which could make it confusing for readers to navigate.
- In experiment 4.2., my understanding is that both the steering manipulation (equation 2, lines 314-315) and the method to evaluate log-odds (using the output of a linear probe) are linear, so it seems tautological that any features will be "disjoint" here? What am I missing?
- the steering experiment (4.1.) doesn't really involve the usual meaning of "steering" as in "generating text from an LLM while applying a steering vector to activations during the forward pass", but instead is (somewhat loosely) inspired by the idea (because of the use of the last-layer probe and attribution back to the SAE's space). Ideally, there would be a sampling-based evaluation that tells us how well steering with the SAE latents works here.

**Questions:**

- Is the setup in 4.2. not trivial?
- How do you think about bridging the gap from synthetic controlled datasets to more realistic ones?

---

> ### Author Response · Authors · 2025-11-21
> **Response to tsv2**
>
> We appreciate the detailed suggestions for improvement. We have made the following changes to our submission to address these concerns (more detail below):
> * Clarified that we need different techniques to evaluate features that locate concepts, and those that enable steering
> * Clarified the disjointness evaluation
> * Clarified what we mean by steering, and cited work that uses “steering” in a similar sense
> * Added additional training details and example data points in Appendices A and B
>
> > There are lots of different procedures to assign SAE latent(s) to a concept (at least three by my count)
>
> Arad et al. (2025) [1] recently discovered that features that capture a concept in input space are distinct from features that produce that concept in the model’s outputs. More formally, given the concepts that explain the data generation process $z_i \in Z$, latent codes $f_j \in \mathbf{f}$, and outputs corresponding to concepts $y_i \in Y$, is has been found that the features $f_j$ that optimize $p(z_i | f_j)$ are distinct from the features that optimize $p(y_i | f_j)$. This implies that the **MCC evaluations of $p(z_i | f_j)$ and steering-based evaluations of $p(y_i | f_j)$ can be thought of as independent metrics that quantify notions of intrinsic disentanglement (in representation space) and extrinsic disentanglement (utility for steering)**, respectively. When replicating our steering experiments using the top-correlated features, we observed little impact on the model’s output behavior; our data thus empirically supports the existence of this distinction. We have made our motivation for using separate techniques clearer, and will include some qualitative steering results using correlations.
>
> We conceptualize $k$-sparse probing as a multi-dimensional extension of looking for the top-correlated features. This ties in nicely with the completeness and explicitness metrics of DCI-ES; we will clarify this conceptual connection as well.
>
> > Both the steering manipulation (equation 2, lines 314-315) and the method to evaluate log-odds (using the output of a linear probe) are linear, so it seems tautological that any features will be "disjoint" here?
>
> Note that the SAEs are trained on the middle layer, whereas the probe is trained on the final layer (for reasons explained in Footnote 6 in Sec. 4); **non-linearities between layers can mix disjoint features into similar subspaces, so perfect distjointness isn’t guaranteed**. Additionally, two features in the same SAE could map to overlapping subspaces in the original activation space (although this should be rare if the SAEs are trained well).
>
> We observe a few cases where disjointness does not hold; see Fig. 14 in Appendix I.
>
> > The steering experiment (4.1.) doesn't really involve the usual meaning of "steering" as in "generating text from an LLM while applying a steering vector to activations during the forward pass"
>
> Geometrically speaking, these are usually equivalent: modifying an SAE feature’s activation is equivalent to applying a steering vector to activations during the forward pass. One can conceptualize features as weighted combinations of neuron activations; thus, changing a feature will change a specific direction in activation space, similar to how steering vectors modify activations.
>
> This notion of steering as affecting downstream representations rather than generations is also used in [2,3], who validated the effect of steering on probes. To verify that steering causes the model to change its outputs in the expected manner, we have generated textual outputs before and after steering, and added them to the new Appendix K. We observe that steering with the top features appears to be working as intended upon qualitative inspection.
>
> > Only a synthetic dataset evaluation
>
> We now acknowledge this as a limitation in the main text. Our synthetic sentences use natural language constructions, providing partial ecological validity while maintaining ground-truth labels essential for our controlled evaluation. We agree that adding more concepts would strengthen the claims, but believe that the current evidence is sufficient to support our main claims.
>
> # References
>
> [1] Arad et al. (2025). “SAEs Are Good For Steering - If You Select the Right Features.” EMNLP. https://aclanthology.org/2025.emnlp-main.519/
>
> [2] Brinkmann et al. (2025). “Large Language Models Share Representations of Latent Grammatical Concepts Across Typologically Diverse Languages.” NAACL. https://aclanthology.org/2025.naacl-long.312/
>
> [3] Marks et al. (2025). “Sparse Feature Circuits: Discovering and Editing Interpretable Causal Graphs in Language Models.” ICLR. https://arxiv.org/abs/2403.19647

---

### Author Response · Authors · 2025-11-21
**Global response**

We thank the reviewers for their thoughtful and constructive feedback.  We appreciate that reviewers found the controlled-yet-nontrivial setup interesting, the experiments diverse in scope, and the presentation clear. We have integrated a great deal of this feedback into the manuscript, and believe that this has made the paper’s claims significantly stronger and clearer.

Here, we address concerns shared across reviewers. Reviewers pointed out missing methodological details, including examples from the generated dataset, detail on the CFG-based data generation, and training details such as hyperparameters and final losses. We have thus added two new sections to the appendix:
* **Appendix A** contains details about our dataset and how it is generated. We show snippets of the CFG we use to generate our data, and give examples from the dataset.
* **Appendix B** contains the details needed to reproduce our SAE training setups, and also gives common quantitative measures of SAE quality like MSE and sparsity.

Reviewers also agreed that it was difficult to parse the results of the steering experiments, as most of the conclusions were based on visual inspection. We have **added quantitative metrics** to better summarize the results of steering experiments; see the revised Section 4, and specifically Tables 1 and 2.

We have also **added results for SSAEs**, a principled and relatively new method from the causal representation learning literature, as well as more fine-grained quantitative evaluation of identifiability with the DCI-ES score.

Finally, we have refined the narrative to clarify our contributions. **Our primary goal is to form an empirical bridge between causal representation learning (CRL) and mechanistic interpretability (MI).** To do so, we have focused on the distinction between identifiability (CRL) and steering (MI): importantly, one does not necessarily imply the other.

We would be happy to continue the discussion!

---

> ### Author Response · Authors · 2025-11-27
>
> Dear Reviewers,
>
> Thank you again for your constructive feedback and suggestions to improve our submission. As the rebuttal period is coming to a close, we would like to kindly ask you to let us know if we have addressed all your questions and concerns, or if you have further questions.
>
>
> Thank you in advance,
> The authors

---

### Note · Authors · 2026-01-05

I have read and agree with the venue's withdrawal policy on behalf of myself and my co-authors.